# ON THE CONVERGENCE OF SGD UNDER THE OVER-PARAMETER SETTING

## ABSTRACT

With the improvement of computing power, over-parameterized models get increasingly popular in machine learning. This type of model is usually with a complicated, non-smooth, and non-convex loss function landscape. However, when we train the model, simply using the first-order optimization algorithm like stochastic gradient descent (SGD) could acquire some good results, in both training and testing, albeit that SGD is known to not guarantee convergence for non-smooth and non-convex cases. Theoretically, it was previously proved that in training, SGD converges to the global optimum with probability $1 - \epsilon$, but only for certain models and $\epsilon$ depends on the model complexity. It was also observed that SGD tends to choose a flat minimum, which preserves its training performance in testing. In this paper, we first prove that SGD could iterate to the global optimum almost surely under arbitrary initial value and some mild assumptions on the loss function. Then, we prove that if the learning rate is larger than a value depending on the structure of a global minimum, the probability of converging to this global optimum is zero. Finally, we acquire the asymptotic convergence rate based on the local structure of the global optimum.

## 1 INTRODUCTION

With the improvement of the computing power of computer hardware, an increasing number of over-parameterized models are deployed in the domain of machine learning. One of the most representative and successful models is what we called deep neural network (LeCun et al. (2015); Amodei et al. (2015); Graves et al. (2013); He et al. (2016); Silver et al. (2017)), which has achieved great empirical success in various application areas (Wu et al. (2016); Krizhevsky et al. (2017); Silver et al. (2017); Halla et al. (2022)). Meanwhile, deep neural networks are large in scale and have an optimization landscape that is in general non-smooth and non-convex (Wu et al., 2019; Brutzkus & Globerson, 2017). Training such a model should have been concerning. However, people could usually acquire very good results just through using first-order methods such as stochastic gradient descent (SGD). A large theoretical gap persists in understanding this process. Two main questions arise.

1. Due to the over-parametrization and the highly complex loss landscape of deep neural networks, optimizing the deep networks to the global optimum is likely NP-hard (Brutzkus & Globerson, 2017; Blum & Rivest, 1992). Nevertheless, in practice, simple first-order methods, which does not have a convergence guarantee in the non-smooth and non-convex case (Liu et al., 2022a;b), are capable of finding a global optimum. This happens even more often on the training data (Zhang et al., 2021; Brutzkus & Globerson, 2017; Wu et al., 2019). It has been an open problem (Goodfellow et al., 2014) that, in this case, does SGD provably find the global optimum? Does the result generalize to more general model structures beyond neural networks?

2. In general, over-parametrized models offer many global optimums. These global optimums have the same training loss of zero, and meanwhile drastically different test performance (Wu et al., 2018; Feng & Tu, 2021). Interestingly, studies find that SGD tends to converge to those generalizable ones (Zhang et al., 2021). In fact, it is observed empirically that SGD could usually find flat minima, which subsequently enjoys better generalization (Kramers, 1940; Dziugaite & Roy, 2017; Arpit et al., 2017; Kleinberg et al., 2018; Hochreiter & Schmidhuber, 1997; 1994). Why and how does SGD find a flat global minimum? The empirical finding has yet to be theoretically validated.

**Related Works** For the first question, in recent years, there have been a number of theoretical results that target to explain this phenomenon. Many of them focus on concrete neural network models, like two-layer networks with linear active function (Bartlett et al., 2018; Hardt & Ma, 2016). Several works need the inputs to be random Gaussian variables (Ge et al., 2018; Tian, 2017; Du et al., 2017; Zhong et al., 2017). Authors in Wu et al. (2019); Allen-Zhu et al. (2019) consider the non-smooth case, but its techniques is depending on the structure of the network. They prove when the number of nodes is enough large, the objective is "almost convex" and "semi-smooth". The techniques unfortunately do not generalize to more general models. Another commonly used technique is to ignore the non-smoothness and apply the chain rule anyway on the non-smooth points (Bartlett et al., 2018). The derivation does provide some intuitions but they do not offer any rigorous guarantees, as the chain rule does not hold (Liu et al., 2022a;b). Even with these kinds of restrictions, existing works (Ge et al., 2018; Tian, 2017; Du et al., 2017; Bartlett et al., 2018; Vaswani et al., 2019; Chizat & Bach, 2018) only manage to find a high probability convergence result to the global optimum. The difference between this probability and 1 could depend on the structure of the model, like the number of nodes in the neural network, which raises further concerns on the tightness of the probability bound. It is currently lacking to analyze SGD for general models to obtain an almost surely convergence to the global optimum.

For the second question, most works investigate the flat minima in a qualitative way. A recent work is by Xie et al. (2020), which views the SGD process as a stochastic differential equation (SDE), and uses SDE to describe the process of the iteration escaping from the sharp minimum. Similar techniques are also used in the works by Wu et al. (2019); Feng & Tu (2021). Unfortunately, SGD can be viewed as an SDE only when the learning rate is sufficiently small, and for a normal learning rate trajectories formed by SGD and SDE could be arbitrarily different. Another technique used to study this problem is to use the linear stability (Wu et al., 2018; Feng & Tu, 2021), which considers a linear system near a global minimum. The behavior of SGD near some global minimum can then be characterized by the linear system of this global minimum. However, different from a deterministic system where the property near one point can be quantitative determine by the linearized system of this point, a stochastic system property near one point is determined by all points in $\mathbb{R}^d$. Using this linearized function to fully represent SGD near some global minimum is thus not a rigorous argument.

**Contributions**

1. Under several mild assumptions about the non-smooth and non-convex loss function, we provide the first proof that from an arbitrary initialization SGD could make the iteration converge to the global optimum almost surely, i.e., $P(\theta_n$ converges to a global optimum$) = 1$.

2. Under the same set of assumptions and the same setting of SGD, we prove that if the learning rate is larger than a threshold, which depends on the sharpness of a global minimum, the probability which the iteration converges to this global optimum is strictly 0.

3. With similar assumptions and the same setting, we acquire the asymptotic convergence rate of the iteration converging to the global optimum. By this result, we know that SGD achieves an arbitrary accuracy in polynomial time.

**Technical Insight** The basic intuition is as follows. We first understand the SGD as a Markov chain with the continuous state space. Then we aim to prove that the global optimum is the only absorbing state of this Markov chain. Concretely, due to the property of the sampling noise, this noise enjoys 0 variance when the optimization variable $\theta$ reaches the global optimum (Claim 2.1), i.e., $\mathbb{E}_{\xi_n} \|\tilde{\nabla} g(\theta, \xi_n) - \tilde{\nabla} g(\theta)\|^2 = 0$ (notations are defined in the next section), which guarantees that once $\theta_n$ reaches the global optimum, it will not escape from the optimum. Meanwhile, in other local optimums, the positive variance makes $\theta_n$ jump out to this local optimum. Otherwise, as this Markov chain is a continuous state space Markov chain, an absorbing state with the measure 0 cannot become the real absorbing state (the probability of the $\theta_n$ reaching this absorbing state in every epoch is 0). Based on this, we need this absorbing state to have a flat-enough neighborhood (Assumption 2.2 in the new version), which deduces that $\theta_n$ that fall on this neighborhood tend to move closer to this absorbing state. Combining this absorbing state and this neighborhood statement, we can prove the distribution of $\theta_n$ will concentrate on the global optimum when as the iteration goes. Finally, this distribution will degenerate to the global optimum, that is, $\theta_n$ will converge to the global optimum.

This neighborhood is the key insight of proving the convergence of SGD. The neighborhood cannot be very sharp (have at most quadratic growth), which is the reason we made Assumption 2.2, item 1. It is actually reflected in Equation (8). A flat enough neighborhood can make the coefficient of the third term of (8) negative, which in turn makes the $R(\theta_n)$ (the Lyapunov function) to decrease with high probability ($\theta_n$ close to global optimum). Otherwise, if the neighborhood is sharp, this coefficient will become positive, which makes $R(\theta_n)$ increasing ($\theta_n$ away from global optimum).

## 2 PROBLEM FORMULATION

We investigate SGD under the over-parametrization setting, under a few mild assumptions on the objective function. The setting and the assumptions, as well as some preliminaries that are relevant to the results, are provided in Section 2.1. We then present the sampling schemes in Section 2.2.

### 2.1 OPTIMIZATION UNDER OVER-PARAMETRIZATION

In this paper, given a dataset $\mathcal{D} = \{(x_i, y_i)\}$, $x_i, y_i \in \mathbb{R}^d$, we consider a model $\hat{y}_i = f(\theta, x_i)$, and the mean-square error (MSE) loss, i.e.,

$$g(\theta) = \frac{1}{N} \sum_{i=1}^{N} g(\theta, x_i), \ \ g(\theta, x_i) = \big(f(\theta, x_i) - y_i\big)^2. \tag{1}$$

The goal of an optimization method, like SGD, is to obtain an optimum $\theta \in J^*$, where $J^* = \arg\min_{\theta \in \mathbb{R}^d} g(\theta)$.

In the over-parametrization setting, this optimum is zero. To handle the non-smoothness, we recall the definition of Clarke subdifferential (Clarke, 1990), which is an important tool to design and operate SGD algorithms.

**Definition 1** (Clarke subdifferential (Clarke, 1990))**.** *Let $\bar{x} \in \Omega$ be given. The Clarke subdifferential of $f$ at $\bar{x}$ is defined by*

$$\partial f(\bar{x}) = \operatorname{co}\left\{\lim_{x \to \bar{x}} \nabla f(x) : f \text{ is smooth at } x\right\},$$

*where* co *represents the convex hull. If $f$ is furthermore smooth, it holds that $\partial f(x) = \{\nabla f(x)\}$. We use $\tilde{\nabla} f(x)$ to denote an arbitrary element in $\partial f(x)$, and for convenient, we call $\tilde{\nabla}$ as subgradient.*

The Clarke subdifferential does not enjoy the chain rule and several techniques involved in regular gradient cannot be reused in our case. We provide a counterexample to illustrate this in Claim A.1.

This property and a few assumptions to eliminate pathological cases are described in the below assumption.

**Assumption 2.1.** *The loss function $g(\theta)$ satisfies the following conditions:*

1. *$g(\theta)$ is continuous and smooth almost everywhere;*

2. *The global optimum value of $g(\theta)$ is $0$;*

3. *The set of global optimum points $J^*$ is composed of countably connected components $J_i$, i.e., $J^* = \bigcup_{i=1}^{+\infty} J_i$ $(J_i \cap J_j = \emptyset)$;*

4. *There is a scalar $c > 0$, such that whenever $g$ is smooth on $\theta_1, \theta_2$ then for any data point $(x_i, y_i)$,*
   $$\left\|\tilde{\nabla} g(\theta_1, x_i) - \tilde{\nabla} g(\theta_2, x_i)\right\| \le c \max\{\|\theta_1 - \theta_2\|, 1\}.$$

This assumption describes the overall structure of the loss function $g(\theta)$. All 4 items in this Assumption are quite mild and are commonly used in optimization and learning.

Items 1 and 2 are true under the MSE loss and the over-parametrization setting. Item 3 describes that the optimum is composed of countably many connected components and this item holds for almost all functions unless one delicately constructs a pathological counterexample Jin et al. (2022). In

this paper, to make the presentation clear, we continue with the countably many points assumption $J^* = \bigcup_{i=1}^{+\infty} \{\theta_i^*\}$ to avoid the tedious arguments on continuum of optimums. Item 4 can be seen as a non-smooth extension of the traditional $L$-smooth condition, i.e., $\|\nabla g(x) - \nabla g(y)\| \leq L\|x - y\|$. It can be satisfied by many non-smooth functions, like ReLU and leaky-ReLU.

Similar to the regular gradient, the subgradient is also zero at the optimum.

**Claim 2.1.** *For the MSE loss function (1), if the global optimum is* 0, *i.e.,* $\min_{\theta \in \mathbb{R}^d} g(\theta) = 0$. *Then the subgradient at the optimum points $J^*$ is* 0.

*Proof.* For any $\theta_0 \in \{\theta \mid g \text{ is smooth at } \theta\}$, we can get that

$$\tilde{\nabla} g(\theta_0) = \nabla g(\theta_0) = \frac{1}{N} \sum_{i=1}^{N} \left( f(\theta_0, x_i) - y_i \right) \nabla f(\theta_0, x_i).$$

Then for any $\theta^* \in J^*$, we have

$$\lim_{\theta_0 \to \theta^*} \tilde{\nabla} g(\theta_0) = \lim_{\theta_0 \to \theta^*} \frac{1}{N} \sum_{i=1}^{N} \left( f(\theta_0, x_i) - y_i \right) \nabla f(\theta_0, x_i) = 0,$$

where $g$ is smooth at $\theta_0$. Then,

$$\partial g(\theta^*) = \text{co} \left\{ \lim_{\theta_0 \to \theta^*} \nabla g(\theta_0) : f \text{ is smooth at } z \right\} = \text{co}\{0\}.$$

This concludes that $\tilde{\nabla} g(\theta^*) = 0$. $\qquad\square$

Notice that despite that $g$ is non-smooth in general, in our setting, it is smooth on the optimum as described in the above claim. This distinguishes our setting from the line of literature on non-smooth optimization.

To make a global convergence, we need at least one $\theta^* \in J^*$ to be not very "sharp". That is, at the $\delta_{\theta^*}-$ neighboring of $\theta^*$ the loss function holds L-smooth condition with the coefficient $\beta_{\theta^*}$ and an assumption as follow:

**Assumption 2.2.** *There exist $\theta^* \in J^*$, $r_{\theta^*} \geq 1$, $\delta > 0$, a neighboring area $U(\theta^*, \delta_{\theta^*})$ of $\theta^*$, such that for those $\theta \in U(\theta^*, \delta_{\theta^*})$ that $\tilde{\nabla} g(\theta)$ holds*

1. *For any mini-batch $C_i$, $g_{C_i}(\theta)$ holds the local one point L-smooth condition, i.e. $\|\tilde{\nabla} g(\theta)\| < \beta_{\theta^*}\|\theta - \theta^*\|$ ($\forall\, \theta \in U(\theta^*, \delta_{\theta^*})$).*

2. *The loss function holds $\tilde{\nabla} g(\theta)^T (\theta - \theta^*) > \alpha_{\theta^*}\|\theta - \theta^*\|^{r_{\theta^*}+1}$ ($\forall \theta \in U(\theta^*, \delta_{\theta^*})$), for some constant $\alpha_{\theta^*} > 0$.*

The first item of this assumption is very mild. Due to Claim 2.1, we know $g(\theta)$ is smooth in $\theta^*$, that is, $\lim_{\theta \to \theta^*} \tilde{\nabla} g(\theta) = \nabla g(\theta^*) = 0$. Then item 1 is just to bound the speed of subgradient tend to 0 is not slower than a linear function (not too sharp as $O(\sqrt{\|\theta - \theta^*\|})$ or $O(\|\theta - \theta^*\|^{0.9})$). The second item of this assumption is very close to the local Kurdyka-Lojasiewicz condition, i.e. $\|\nabla g(\theta)\|^{2r} \geq g(\theta) - g(\theta^*)$ $(r \geq 1)(\theta \in U(\theta^*, \delta_{\theta^*}))$ which is a typically mild condition used to substitute the local Polyak-Łojasiewicz condition (item 2 and the local Kurdyka-Lojasiewicz condition are totally equivalent for an unary function). This assumption is milder than several assumptions used in the previous works. It can be seen as the loss function has an $r_{\theta^*} + 1$-order Taylor expansion on $\theta^*$. Compared with the one point strongly convexity used in Li & Yuan (2017); Kleinberg et al. (2018), the positive Hessian matrix and local Polyak-Łojasiewicz condition in global optimum used in Wu et al. (2018); Jin et al. (2022), our assumption is much milder.

## 2.2 TWO TYPES OF NOISE OF SGD

In the rest of this section we describe two types of SGD algorithms, by different sampling noise. The first type is with the traditional sampling noise while the second type is SGD with the sampling noise with global stable guarantee. They involve slightly different assumptions and the analysis of SGD also varies by the type of noise. Nevertheless, they conclude similar results as we will present in the next section.

### 2.2.1 REGULAR SAMPLING NOISE

We start with the iterations of an (regular) SGD algorithm, that

$$
\begin{aligned}
v_n &= \epsilon_0 \tilde{\nabla} g(\theta_n, \xi_n)\,, \\
\theta_{n+1} &= \theta_n - v_n\,,
\end{aligned}
\tag{2}
$$

where $\{\xi_n\}$ represents the sampling noise. That is, we have the noised sampling

$$
\tilde{\nabla} g(\theta, \xi_n) = \frac{1}{|C_i|} \sum_{\bar{x},\,\bar{y} \in C_i} \tilde{\nabla}\Big( \big( f(\theta, \bar{x}) - \bar{y} \big)^2 \Big)\,,
$$

where $C_i$ is a randomly selected mini-batch from the original data set. The next statement assumes that the subdgradient can be sampled without the sampling error being too large. It is necessary for an algorithm to use the gradient:

**Assumption 2.3.** *Let $\xi_n$ be the sampling noise involved in the $n$-th iteration of SGD and $\tilde{\nabla} g(\theta, \xi_n)$ be the noised sampling of the subgradient. For any $\theta \in \mathbb{R}^d$, it holds*

$$
\liminf_{\theta \to \infty} \| \nabla g(\theta) \| > 0,
$$

*and*

$$
\limsup_{\theta \to +\infty} \frac{\mathbb{E}_{\xi_n} \big\| \tilde{\nabla} g(\theta, \xi_n) \big\|^2}{\big\| \tilde{\nabla} g(\theta) \big\|^2} < M_0,
$$

*where $M_0 \geq 0$ is a constants decided by $g$. Meanwhile, we need $\liminf_{\theta \to \infty} \| \tilde{\nabla} g(\theta) \| > \max\{ 4c\sqrt{M_0}, 4c\sqrt{K_0} \}$.*

First of this assumption is milder than the widely used *bounded variance assumption*, i.e., $\mathbb{E}_{\xi_n} \big\| \tilde{\nabla} g(\theta, \xi_n) - \tilde{\nabla} g(\theta) \big\|^2 \leq a$ (Li & Yuan, 2017; Kleinberg et al., 2018). Second part is to combine the $\{\theta_n\}$ tend to $\infty$. For example, for a very simple loss functions $g(\theta) = \frac{1}{3}\big( \|\theta - \theta_1\|^2 + \|\theta - \theta_2\|^2 + \|\theta - \theta_3\|^2 \big)$, It hold our Assumption 2.3 but not hold bounded variance assumption. Meanwhile, this sampling immediately implies the below bound.

**Claim 2.2.** *For any bounded set $Q$ that include $J^*$, it holds*

$$
\mathbb{E}_{\xi_n} \big\| \tilde{\nabla} g(\theta, \xi_n) \big\|^2 \leq G_Q g(\theta) \ (\forall\, \theta \in Q)\,,
$$

*where $G_Q$ is a constant decided by $Q$.*

*Proof.* For any smooth point in $Q$, the mini-batch gradient norm satisfies

$$
\begin{aligned}
\big\| \tilde{\nabla} g_{C_i}(\theta) \big\|^2 &= \frac{4}{|N_0|^2} \left\| \sum_{x_c \in C_i}^{N} \big( f(\theta, x_c) - y_c \big) \tilde{\nabla} f(\theta, x_c) \right\|^2 \\
&\leq \frac{4}{|N_0|^2} \sum_{x_c \in C_i}^{N} \big( f(\theta, x_c) - y_c \big)^2 \big\| \tilde{\nabla} f(\theta, x_c) \big\|^2 \leq \frac{4N \sum_{i=1}^{N} \big\| \tilde{\nabla} f(\theta, x_i) \big\|^2}{N_0^2} g(\theta),
\end{aligned}
\tag{3}
$$

where $N_0$ is the size of the mini-batch. Define

$$
h_{C_i}(\theta) = \frac{4N \sum_{i=1}^{N} \big\| \tilde{\nabla} f(\theta, x_i) \big\|^2}{N_0^2}\,.
$$

Through Assumption 2.1, we know that $h(\theta)$ is bounded on smooth points. Then we have

$$
\big\| \tilde{\nabla} g_{C_i}(\theta, \xi_n) \big\|^2 \leq \frac{4N \overline{G}_Q}{N_0^2} g(\theta) \ \ (\text{when } g \text{ is smooth at } \theta)\,.
\tag{4}
$$

Then,

$$
\mathbb{E}_{\xi_n} \big\| \tilde{\nabla} g(\theta, \xi_n) \big\|^2 = \frac{C_{N-1}^{N_0-1}}{C_N^{N_0}} \sum_{all\ C_i} \big\| \tilde{\nabla} g_{C_i}(\theta) \big\|^2 \leq \frac{4N \overline{G}_Q C_{N-1}^{N_0-1}}{N_0^2 C_N^{N_0}} g(\theta) := G_Q g(\theta)\,.
$$

For the non-smooth point $\theta$, we can prove for any sequence $\theta_0 \to \theta$ ($g$ is smooth at $\theta_0$), through Equation (4), there is

$$\left\| \lim_{\theta_0 \to \theta} \tilde{\nabla} g_{C_i}(\theta_0, \xi_n) \right\|^2 = \lim_{\theta_0 \to \theta} \left\| \tilde{\nabla} g_{C_i}(\theta_0, \xi_n) \right\|^2 \leq \frac{4N\overline{G}_Q}{N_0^2} \lim_{\theta_0 \to \theta} g(\theta_0) = \frac{4N\overline{G}_Q}{N_0^2} g(\theta) \,.$$

Recall the following fact:

If $\|a_1\|^2 < s_0$, $\|a_2\|^2 < s_0$, $\ldots$, $\|a_n\|^2 < s_0$, the norm of their any convex combination

$$\|\bar{a}\|^2 := \left\| \sum_{i=1}^n \lambda_i a_i \right\|^2 < \left( \sum_{i=1}^n \lambda_i^2 \right) s_0 \leq s_0 \,.$$

Then we obtain

$$\left\| \tilde{\nabla} g_{C_i}(\theta) \right\|^2 \leq \frac{4N\overline{G}_Q}{N_0^2} g(\theta) \,.$$

This concludes that

$$\mathbb{E}_{\xi_n} \left\| \tilde{\nabla} g(\theta, \xi_n) \right\|^2 \leq G_Q g(\theta) \,. \qquad \qquad \square$$

We could observe that the noise variance $\mathbb{E}_{\xi_n} \|\tilde{\nabla} g(\theta, \xi_n) - \tilde{\nabla} g(\theta)\|^2 = 0$ at the global optimum (Claim 2.1). Intuitively, the zero variance makes the $\theta_n$ stable in the global optimum, while for a local minimum or a saddle point the variance is nonzero in general. This is intuitively how SGD escapes from local minimum and saddle points.

We have to notice that the global optimum is a subset of the set where the noise variance equals $0$. It is easy to prove that

$$J^* \subseteq \left\{ \theta \mid \mathbb{E}_{\xi_n} \|\tilde{\nabla}(\theta, \xi_n) - \tilde{\nabla}(\theta)\|^2 = 0 \right\} = J^{**} \,,$$

where $J^{**}$ is equivalent to

$$J^{**} = \bigcap_{C_i} \left\{ \theta \mid \tilde{\nabla} \left( \left( f(\theta, \bar{x}) - \bar{y} \right)^2 \right) = 0 \right\} \,.$$

Our techniques will eventually prove that the SGD with regular sampling noise converges to $J^{**}$. This could be different than $J^*$ in theory, but intuitively, for the over-parameter model and a large amount of data the model $f(\theta, x)$ is complex enough to make sure that other stationary points are sensitive to the mini-batch batch selection. As such making a point, that is not the global optimum, stationary to all batches simultaneously is almost impossible, i.e., $J^{**}/J^* = \emptyset$. Nevertheless, in order to insure the rigor of the theory, we make an additional assumption only for the regular sampling noise. This assumption is lifted in the sampling noise with global stable guarantee.

**Additional assumption for regular sampling noise** *For the sampling noise $\{\xi_n\}$, points that are stationary to all mini-batches must be in $J^*$, i.e., $J^* = J^{**}$. Meanwhile, for every mini-batch loss function $g_{C_i}$, the stationary point set of $g_{C_i}$ is countable.*

If one slightly modifies SGD by adding an additional Gaussian noise, we will prove that such sampling noise will enjoy a global stable guarantee. With this variant of SGD, the above assumption could be lifted. We now present our proposed variant of SGD.

### 2.2.2 Sampling noise with global stable guarantee

The sampling noise we propose in this section is the regular noise in SGD plus an extra Gaussian noise, as

$$\begin{aligned} v_n &= \epsilon_0 \left( \tilde{\nabla} g(\theta_n, \xi_n) + \sqrt{\min\{g(\theta_n), K_0\}} \tau_n \mathcal{N}_n \right) , \\ \theta_{n+1} &= \theta_n - v_n \,, \end{aligned} \tag{5}$$

where $\{\xi_n\}$ again represents the sampling noise, $K_0$ is a constant to prevent the noise from approaching infinity, $\{\mathcal{N}_n\}$ represents a mutually independent standard Gaussian noise, $\{\tau_n\}$ is a mutually independent Bernoulli variable, i.e., $P(\tau_n = 0) = p_0$, $P(\tau_n = 1) = 1 - p_0$, and $\{\tau_n\}, \{\xi_n\}, \{\mathcal{N}_n\}$ are also mutually independent. The coefficient $\min\{g(\theta_n), K_0\}$ is to make sure the algorithm hold a positive noise variance $\mathbb{E}_{\xi_n, \tau_n, \mathcal{N}_n} \|v_n\|^2 > 0$ in non-optimal stationary points. We use $\{\tau_n\}$ to reduce the scale of the problem, making the scale of the new noise equal to the scale of the mini-batch

gradient $\tilde{\nabla} g(\theta_n, \xi_n)$ and as the original sampling noise. For example, if the batch size is 100 and the scale of the original data set is 10000, then we can set $p_0 = 1 - 0.01$, which makes the average scale of the noise $\min\{g(\theta_n), K_0\}\tau_n\mathcal{N}_n)$ also 100. The tail term $\sqrt{\min\{g(\theta_n), K_0\}\tau_n\mathcal{N}_n)}$ guarantees that this algorithm has a positive variance in $\mathbb{R}^d/J^*$.

## 3 MAIN RESULTS

Our first main result states that SGD must converge to a global optimum with probability 1. This is a large improvement from previous results with only $1 - \delta$ probability, where $\delta$ depends on the model. Our theorem answers the question raised in the introduction, affirmatively, that SGD could indeed obtain a global optimum even in this non-smooth non-convex over-parameter setting. The next two theorems discuss the cases of $r_{\theta^*} > 1$ (higher than second-order local structure) and $r_{\theta^*} = 1$ (second-order local structure) respectively.

**Theorem 3.1.** *Consider the SGD iteration in Equation (5), or alternatively Equation (2) with $J^* = J^{**}$, and the MSE loss function (1). If Assumptions 2.1, 2.3 hold, and Assumption 2.2 holds with $r_{\theta^*} > 1$, then for any $0 < \epsilon_0 < \min\{1/2cM_0, 1/4cK_0(1-p_0)\}$, and for any initialization $\theta_1 \in \mathbb{R}^d$, $\{\theta_n\}$ converges to the set $J^*$ almost surely, i.e.,*

$$\lim_{n\to\infty} d(\theta_n, J^*) = 0 \quad a.s.,$$

*where $d(x, J^*) = \inf_y\{\|x - y\|, y \in J^*\}$ denotes the distance between point $x$ and set $J^*$. Meanwhile the value of the loss function converges to $0$ almost surely, i.e.,*

$$\lim_{n\to\infty} g(\theta_n) = 0 \quad a.s..$$

For each main result, we provide a proof sketch to illustrate our idea in deriving the result. A rigorous argument is deferred to the appendix.

*Proof sketch.* Our proof mainly relies on two techniques. The first technique is the Lyapunov method. It transfers the convergence of a high dimension vector $\theta_n$ to a one dimensional Lyapunov function $R(\theta_n)$. The second technique is to use the idea of Markov process. We sketch these two steps and an additional step as follows.

**Step 1**: In this step, we aim to prove that there exists at least one bounded set $S_0$ such that there is no limit point of $\{\theta_n\}$ is in it almost surely. Through the Borel–Cantelli Lemma, it amounts to proving

$$\sum_{n=1}^{+\infty} P(\theta_n \in S_0) < +\infty. \tag{6}$$

In order to prove Equation (6), we use the Lyapunov method, constructing a Lyapunov function $R(\theta)$ which holds a unique zero $R(\theta^*) = 0$ and an open set $\hat{S}_0$ which include $\theta^*$ (exact forms of $R(\theta)$ and $\hat{S}_0$ are provided in the appendix). We assign $I_n$ as the characteristic function of the event $\{\theta_n \in \hat{S}_0\}$. Then we obtain the inequality

$$I_{n+1}^{(\hat{S}_0)} R(\theta_{n+1}) - I_n^{(\hat{S}_0)} R(\theta_n) \leq -I_n^{(\hat{S}_0)} R^{\frac{2r}{r+1}}(\theta_n) + u_n, \tag{7}$$

where $u_n$ is defined in (12) with $\sum_{n=1}^{+\infty} \mathbb{E}(u_n) < +\infty$. Summing up Equation (6) yields

$$\sum_{n=1}^{+\infty} \mathbb{E}\left(I_n^{(\hat{S}_0)} R^{\frac{2r}{r+1}}(\theta_n)\right) < \mathbb{E}\left(I_1^{(\hat{S}_0)} R^{\frac{2r}{r+1}}(\theta_1)\right) + \sum_{n=1}^{+\infty} \mathbb{E}(u_n) < +\infty.$$

Subsequently we could construct $S_0 := \hat{S}_0/U(\theta^*, \delta_0')$, for some small enough $\delta_0'$, to make $\sum_{n=1}^{+\infty} \mathbb{E}\left(I_n^{(S_0)} R^{\frac{2r}{r+1}}(\theta_n)\right) < \sum_{n=1}^{+\infty} \mathbb{E}\left(I_n^{(\hat{S}_0)} R^{\frac{2r}{r+1}}(\theta_n)\right) < +\infty$. Then, as whenever $\theta_n \in \hat{S}_0/U(\theta^*, \delta_0')$ we have $R^{\frac{2r}{r+1}}(\theta) > \tilde{\epsilon}$, we have

$$\sum_{n=1}^{+\infty} P(\theta_n \in S_0) < \frac{1}{\tilde{\epsilon}} \sum_{n=1}^{+\infty} \mathbb{E}\left(I_n^{(S_0)} R^{\frac{2r}{r+1}}(\theta_n)\right) < +\infty.$$

As such we conclude Equation (6), and through the Borel–Cantelli Lemma, we know that there is no limit point in $S_0$ almost surely.

**Step 2**: In this step, we aim to prove that for any bounded set $S$ that has no intersection with $J^*$, there is no limit point in it. The way we prove it is different for the two types of noise (2) and (5). Handling the sampling noise with global stable guarantee (5) is relatively simple. The Gaussian noise of (5) guarantees that it forms an irreducible Markov process. Then using the property of the irreducible Markov process directly will prove the statement. For (2), the situation becomes complicated where an argument of the regular sampling noise does not deduce an irreducible Markov process. We prove it using a delicate argument. We first prove that a max positive bounded invariant set $D$ must hold its boundary set $\partial D \cap J^* \neq \emptyset$, and every trajectory started from this set must almost surely converge to some global optimum. Here a set is max positive invariant if any trajectories started in $S_0$ will not escape $S_0$ and for any points $\theta'' \notin J^* \cup D$, $\theta'' \cup D$ is not a positive invariant set. That means, for any point either almost every trajectory started with it converges to $J^*$, or it holds a positive probability transfer to $S_0$. For the first situation, this statement is satisfied. For the second situation, we can make a small enough positive measure set, such that for any $\theta \in S$, there exists a $\delta'_0$, and some large enough $k$, $P(\theta_{n+k} \in S_0 \mid \theta_n = \theta) > \upsilon$. Then we can get as desired

$$
\upsilon \sum_{n=1}^{+\infty} P(\theta_n \in S) = \upsilon \sum_{n=k+1}^{+\infty} \int_S P_{n-k}(d\theta) \leq \sum_{n=k+1}^{+\infty} \int_S P(\theta_{n+k} \in S^{(\delta_0, l_0)} \mid \theta_n = \theta) P_{n-k}(d\theta)
$$

$$
= \sum_{n=k+1}^{+\infty} \int_{S^{(\delta_0, l_0)}} P_n(d\theta) < +\infty \, .
$$

**Step 3**: In the previous step we actually proved that almost surely either $\theta_n \to J^*$ or $\theta_n \to +\infty$. Through the Kolmogorov 0-1 law, we know $\{\theta_n \text{ converges}\}$ is a tail event. As such, $P(\theta_n \to J^*) \in \{0, 1\}$. Meanwhile as $P(\theta_n \to \infty) = 1$ is impossible, $P(\theta_n \to J^*)$ could only take 1. □

In step 3, we suspect that $P(\theta_n \to \infty) = 1$ is indeed impossible, even without the assumption $\liminf_{\theta \to \infty} \|\tilde{\nabla} g(\theta)\| > \max\{4c\sqrt{M_0}, 4c\sqrt{K_0}\}$. In fact, as long as $\theta_n$ converges to $J^*$ for any initialization $\theta_1$ in some neighboring domain of the optimum, it converges for all initialization. This is because for every initialization it either converges to the optimum or it has a positive probability to transfer to an arbitrary set with a positive measure. As the neighboring domain could be arbitrarily small, it is likely to exist.

**Theorem 3.2.** *Consider the SGD iteration in Equation (5), or alternatively Equation (2) with $J^* = J^{**}$, and the MSE loss function (1). If Assumptions 2.1, 2.3 hold, and Assumption 2.2 holds with $r_{\theta^*} = 1$, then for any $0 < \epsilon_0 < \min\{1/2cM_0, \alpha_{\theta^*}/2(2 - p_0)\beta_{\theta^*}^2, 1/4cK_0(1 - p_0)\}$, where normal sampling noise 2 can be seen as $p_0 = 0$, and for any $\theta_1 \in \mathbb{R}^d$, $\theta_n$ converges to $J^*$ almost surely, i.e.,*

$$
\lim_{n \to \infty} d(\theta_n, J^*) = 0 \quad a.s. \, ,
$$

*where $d(x, J^*) = \inf_y\{\|x - y\|, y \in J^*\}$ denotes the distance between point $x$ and set $J^*$. Meanwhile the value of the loss function converges to $0$ almost surely, i.e.,*

$$
\lim_{n \to \infty} g(\theta_n) = 0 \quad a.s. \, .
$$

*Proof sketch.* This proof will be similar to the proof of Theorem 3.1. The difference is when $r_{\theta^*} = 1$ the convergence towards a global optimum with second-order local structure is conditional on the selection of the initial learning rate $\epsilon_0$. The reason for this is the inequality

$$
I_{n+1}^{(\hat{S}_0)} R(\theta_{n+1}) - I_n^{(\hat{S}_0)} R(\theta_n) \leq -\left(\alpha_{\theta^*}\epsilon_0 - 2(2 - p_0)\epsilon_0^2\beta_{\theta^*}^2\right) I_n^{(\hat{S}_0)} R(\theta_n) + u_n \tag{8}
$$

holds only when the coefficient $\alpha_{\theta^*}\epsilon_0 - 2(2 - p_0)\epsilon_0^2\beta_{\theta^*}^2 > 0$. By setting $\epsilon_0$ as the theorem the inequality and other arguments remain valid. This proof also agrees with our intuition that SGD converges to a sharper global optimum not as easy as a flat one ($r_{\theta^*} > 1$). □

Recall the second question raised in SGD was conjecturing if SGD tends to choose the flat minima (and so as to enjoy a better generalization). In the end of the above proof we find that SGD converges

to a sharper global optimum not as easy as a flat one. This observation is through positive results only, though. We wonder if the converse is also true, that is, if a global minimum is not flat, then SGD is unlikely to converge to that.

In the below theorem we answer the converse affirmatively. Is is proved that if $\epsilon_0$ is large enough, then the iteration will almost surely not converge to this optimum.

**Theorem 3.3.** *Consider the SGD iteration in Equation (5), or alternatively Equation (2) with $J^* = J^{**}$, and the MSE loss function (1). If Assumptions 2.1, 2.3 hold, and Assumption 2.2 holds with $r_{\theta^*} = 1$, then for any $\theta_1 \in \mathbb{R}^d$, if $\epsilon_0 > \beta_{\theta^*}/2(2 - p_0)\alpha_{\theta^*}^2$, where normal sampling noise 2 can be seen as $p_0 = 0$, the probability that $\theta_n$ converges to $\theta^*$ is 0, i.e.,*

$$P\Big( \lim_{n \to \infty} \|\theta_n - \theta^*\| = 0 \Big) = 0 \,.$$

*Proof sketch.* The main idea is to prove that if the iteration always stays in a neighboring domain of $\theta^*$, then the probability that this iteration converges to $\theta^*$ is zero. The Lyapunov method is helpful in this case.

**Step 1**: In this step, we aim to acquire a reverse inequality of (7). We first construct a Lyapunov function $R(\theta)$ and a domain $S_1$ of $\theta^*$, and an event $A_{i,n} = \{\theta_{n_0} \in S_1, \ n_0 \in [i, n]\}$ as well its characteristic function $I_{i,n}$. Then we can acquire an inequality

$$I_{i,n}\big(R(\theta_{n+1}) - R(\theta_n)\big) \ge \big(2(2 - p_0)\epsilon_0^2\alpha_{\theta^*}^2 - \epsilon_0\beta_{\theta^*}\big)I_{i,n}R(\theta_n) + I_{i,n}\zeta_n \,, \tag{9}$$

where $\zeta_n$ is defined by equation 33. Notice that if $\big(2(2-p_0)\epsilon_0^2\alpha_{\theta^*}^2 - \epsilon_0\beta_{\theta^*}\big) > 0$, then this inequality will be a variant of diffusion process.

**Step 2**: In this step, we aim to prove when $n$ approaches infinity, the iteration will transform a fixed part of itself out of $S_1$. Through (9), we get

$$\mathbb{E}\big(I_{i,n+1}R(\theta_{n+1})\big) \ge \left(1 + \hat{p}_0 - \frac{\mathbb{E}\big(R(\theta_{n+1})(I_{i,n} - I_{i,n+1})\big)}{\mathbb{E}\big(I_{i,n}R(\theta_n)\big)}\right)\mathbb{E}\big(I_{i,n}R(\theta_n)\big) \,.$$

We know if

$$\frac{\mathbb{E}\big(R(\theta_{n+1})(I_{i,n} - I_{i,n+1})\big)}{\mathbb{E}\big(I_{i,n}R(\theta_n)\big)} < \hat{p}_0 \,,$$

then $\mathbb{E}\big(I_{i,n+1}R(\theta_{n+1})\big)$ will diverge to infinity, which is impossible to happen. As such, it must hold

$$\frac{\mathbb{E}\big(R(\theta_{n+1})(I_{i,n} - I_{i,n+1})\big)}{\mathbb{E}\big(I_{i,n}R(\theta_n)\big)} \ge \hat{p}_0 \,.$$

**Step 3**: In this step, we will show that if $\mathbb{E}(I_{i,+\infty} \neq 0) > 0$, then $I_{i,n}R(\theta_n)$ will not converge to 0 almost surely. We prove it by contradiction and assume $P\big(\lim_{n \to +\infty} I_{i,n}R(\theta_n) = 0\big) = 1$. That means for any $\epsilon_0' > 0$, $P\Big(I_{i,n}R(\theta_n) > \epsilon_0'\Big) \to 0$, which concludes $P\Big(I_{i,n}R(\theta_n) \le \epsilon_0'\Big) \to 1$. Then

$$\frac{\mathbb{E}\big(R(\theta_{n+1})(I_{i,n} - I_{i,n+1})\big)}{\mathbb{E}\big(I_{i,n}R(\theta_n)\big)} \to k'\epsilon_0' \,.$$

This forms a contradiction.

**Step 4**: In this final step, we will prove $P\big(\lim_{n \to +\infty} \theta_n = \theta^*\big) = 0$. We inspect the event $\{\theta_n \to \theta^*\}$. If $\mathbb{E}(I_{i,+\infty} \neq 0) > 0$, then due to $\lim_{n \to +\infty} I_{i,n}g(\theta_n) - I_{i,+\infty}g(\theta_n) = 0 \ a.s.$, we could get $P\big(\lim_{n \to +\infty} I_{i,+\infty}R(\theta_n) = 0\big) = 0$. Then,

$$P\big(\{\theta_n \to \theta^*\} \cap A_{i,+\infty}\big) = P\big(\lim_{n \to +\infty} I_{i,+\infty}R(\theta_n) = 0\big) = 0 \,.$$

Otherwise if $\mathbb{E}(I_{i,+\infty} \neq 0) = 0$, we have

$$P\big(\{\theta_n \to \theta^*\} \cap A_{i,+\infty}\big) \le \mathbb{E}(I_{i,+\infty} \neq 0) = 0 \,.$$

Absolutely, we have

$$\{\theta_n \to \theta^*\} \subset \left\{ \bigcup_{i=1}^{+\infty} A_{i,+\infty} \right\}.$$

Subsequently we have

$$P(\theta_n \to \theta^*) = P\left( \{\theta_n \to \theta^*\} \bigcap \left\{ \bigcup_{m=1}^{+\infty} A_{i,+\infty} \right\} \right) = P\left( \bigcup_{i=1}^{+\infty} \{\theta_n \to \theta^*\} \bigcap A_{i,+\infty} \right)$$

$$\le \sum_{i=1}^{+\infty} P\left( \{\theta_n \to \theta^*\} \bigcap A_{i,+\infty} \right) = 0. \qquad \square$$

As we have shown the asymptotic convergence of SGD, the natural question is how fast it converges. To provide the convergence rate, we will need a slightly stronger version of Assumption 2.2. We need, instead of just one $\theta^*$, all $\theta^*$, to satisfy the order $r + 1$ expansion. In this case, the supremum of the expansion order, among all optimum points, is denoted as $\hat{r} = \max_{\theta^* \in J_\infty^*} r_{\theta^*}$, where $J_\infty^* := \{\theta^* \in J^* \mid P(\theta_n \to \theta^*) > 0\}$.

Our next theorem provides the convergence rate of SGD.

**Theorem 3.4.** *Consider the SGD iteration in Equation (5), or alternatively Equation (2) with $J^* = J^{**}$, and the MSE loss function (1). If Assumptions 2.1, 2.3 hold, and the variant of Assumption 2.2 described immediately preceding this statement holds with order $\hat{r} + 1$, then for any $\theta_1 \in \mathbb{R}^d$, $\theta_n$ has an asymptotic convergence rate as*

$$g(\theta_n) = \begin{cases} O(p_0^n) & a.s., & \text{if } \hat{r} = 1, \\ O(n^{-\frac{2}{\hat{r}-1}}) & a.s., & \text{if } \hat{r} > 1, \end{cases}$$

*where $p_0 < 1$ is a constant decided by the learning rate $\epsilon_0$.*

*Proof sketch.* The proof of this theorem is based on the proof of Theorem 3.1. We asymptotically bound of martingale difference (Lemma A.1) and with the bound apply the martingale convergence theorem. The asymptotic convergence rate follows. $\qquad \square$

As an immediate consequence of the convergence rate, the SGD algorithm could obtain an arbitrary accuracy in polynomial time. This validates the efficiency of SGD.

**Corollary 3.1.** *Consider the same setting as Theorem 3.4. For any $\theta_1 \in \mathbb{R}^d$, the computational time for $g(\theta_n)$ to reach an $\eta$ accuracy is*

$$\begin{cases} O\left(N_0 d \cdot \log(\frac{1}{\eta})\right) & a.s., & \text{if } \hat{r} = 1, \\ O\left(N_0 d \cdot (\frac{1}{\eta})^{\frac{\hat{r}-1}{2}}\right) & a.s., & \text{if } \hat{r} > 1, \end{cases}$$

*where $N_0$ is the mini-batch size.*

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

# A APPENDIX

## A.1 COUNTER-EXAMPLE

**Claim A.1.** *The chain rule does not hold the Clarke subdifferential.*

*Proof.* For a composite nonsmooth function, the chain rule may not hold at the nonsmooth point Liu et al. (2022b). We introduce an example as follows.

Consider

$$\min_{w_1 \in \mathbb{R}, w_2 \in \mathbb{R}, b_1 \in \mathbb{R}, b_2 \in \mathbb{R}} f(w_1, w_2, b_1, b_2)$$
$$:= \left( \left( w_2 \sigma \left( w_1 + b_1 \right) + b_2 \right) + 1 \right)^2 + \left( \left( w_2 \sigma \left( 2w_1 + b_1 \right) + b_2 \right) - 1 \right)^2 . \tag{10}$$

Let $w_2^* = 1$, $b_1^* = 0$, $w_1^* = 0$, $b_2^* = 0$, one can easily see that the SGD method will get stuck at $(w_1^*, w_2^*, b_1^*, b_2^*)$, and

$$\partial f(w_1^*, w_2^*, b_1^*, b_2^*) = \left\{ (t, 0, s, 0)^T : t \in [-4, 2], s \in [-2, 0] \right\},$$
$$f(w_1^* + \epsilon, w_2^*, b_1^*, b_2^*) = 5\epsilon^2 - 2\epsilon + 2 < 2 = f(w_1^*, w_2^*, b_1^*, b_2^*) \text{ for some small positive number } \epsilon .$$

Then, observe that $(w_1^*, w_2^*, b_1^*, b_2^*)$ is neither a local minimizer of equation 10. Moreover, one can see that $(1, 2, -1, -1)$ is a global minimizer of equation 10, at which the function value is 0. $\square$

## A.2 AUXILIARY LEMMAS

**Lemma A.1.** *(Theorem 4.2.13, Lei et al. (2005)) Consider a Martingale difference column $\{X_n, \mathcal{F}_n\}$ that satisfies $\sup_n \mathbb{E}(\|X_{n+1}\|^2 | \mathcal{F}_n) < +\infty$ almost surely. Then it holds that*

$$\sum_{k=1}^n \beta_k X_k = O\left( \sqrt{S_n} \ln^{\frac{1}{2} + \eta}(S_n + e) \right) \text{ almost surely}, \quad \forall \eta > 0,$$

*where $S_n = \sum_{k=1}^n \beta_k^2$.*

**Lemma A.2.** *(Lemma 6 in Jin et al. (2022)) Suppose that $\{X_n\} \in \mathbb{R}^d$ is a non-negative sequence of random variables, then $\sum_{n=0}^\infty X_n < +\infty$ holds almost surely if $\sum_{n=0}^\infty \mathbb{E}\left( X_n \right) < +\infty$.*

**Lemma A.3.** *(Wang et al., 2019) Suppose that $\{X_n\} \in \mathbb{R}^d$ is an $\mathcal{L}_2$ martingale difference sequence, and $(X_n, \mathcal{F}_n)$ is an adaptive process. Then it holds almost surely that $\sum_{k=0}^\infty X_k < +\infty$ if*

$$\sum_{n=1}^\infty \mathbb{E}(\|X_n\|^2) < +\infty, \quad \text{or} \quad \sum_{n=1}^\infty \mathbb{E}\left( \|X_n\|^2 | \mathcal{F}_{n-1} \right) < +\infty,$$

*happens almost surely.*

## A.3 PROOF OF LEMMA A.4.

**Lemma A.4.** *Consider the SGD updates specified in equation 2 (or equation 2 with $J^* = J^{**}$) and the MSE loss function equation 1. If Assumptions 2.1, 2.3 hold, then for any $\epsilon_0 < \min\{1/2cM_0, 1/4cK_0(1 - p_0)\}$, where normal sampling noise 2 can be seen as $p_0 = 0$. Then for any $\theta_1 \in \mathbb{R}^d$, the probability of $\theta_n$ diverge to the infinity is less than 1, i.e., $P(\theta_n \to \infty) < 1$.*

*Proof.* We prove this Lemma by contradiction. We first assume $P(\theta_n \to \infty) = 1$, which means $\theta_n \to \infty$ almost surely. By the Lagrange's mean value theorem, we have

$$g(\theta_{n+1}) - g(\theta_n) = \tilde{\nabla}g(\theta_{\zeta_n})^T(\theta_{n+1} - \theta_n),$$

where $\zeta_n$ is a point between $\theta_n$ and $\theta_{n+1}$. If $\zeta_n$ is a non-smooth point, then we can find at least one point in the set of $\tilde{\nabla}g(\theta_{\zeta_n})$. Therefore, we have

$$
\begin{aligned}
g(\theta_{n+1}) - g(\theta_n) &= \tilde{\nabla}g(\theta_{\zeta_n})^T(\theta_{n+1} - \theta_n) \\
&= -\epsilon_0 \tilde{\nabla}g(\theta_n)^T \tilde{\nabla}g(\theta_n, \xi_n) + \left(\tilde{\nabla}g(\theta_{\zeta_n}) - \tilde{\nabla}g(\theta_n)\right)^T(\theta_{n+1} - \theta_n) \\
&\leq -\epsilon_0 \tilde{\nabla}g(\theta_n)^T \tilde{\nabla}g(\theta_n, \xi_n) + \left\|\tilde{\nabla}g(\theta_{\zeta_n}) - \tilde{\nabla}g(\theta_n)\right\|\|\theta_{n+1} - \theta_n\| \\
&\leq -\epsilon_0 \tilde{\nabla}g(\theta_n)^T \tilde{\nabla}g(\theta_n, \xi_n) + \max\{c, c \cdot \epsilon_0 \|\tilde{\nabla}g(\theta_n, \xi_n)\|\}\epsilon_0 \|\tilde{\nabla}g(\theta_n, \xi_n)\| \\
&< -\epsilon_0 \tilde{\nabla}g(\theta_n)^T \tilde{\nabla}g(\theta_n, \xi_n) + c \cdot \epsilon_0 \|\tilde{\nabla}g(\theta_n, \xi_n)\| + c \cdot \epsilon_0^2 \|\tilde{\nabla}g(\theta_n, \xi_n)\|^2.
\end{aligned}
$$

Through Assumption 2.3, we know that it hold $\mathbb{E}_{\xi_n}\|\tilde{\nabla}g(\theta_n, \xi_n)\|^2 \leq M_0\|\nabla \tilde{g}(\theta_n)\|^2$ when $\theta_n \to \infty$. Then we take an expectation over the sampling noise, we have

$$
\begin{aligned}
\mathbb{E}\left(g(\theta_{n+1})\right) - \mathbb{E}\left(g(\theta_n)\right) <& -\epsilon_0 \mathbb{E}\|\tilde{\nabla}g(\theta_n)\|^2 + c \cdot \epsilon_0\sqrt{M_0}\,\mathbb{E}\|\tilde{\nabla}g(\theta_n)\| + c \cdot M_0 \cdot \epsilon_0^2 \mathbb{E}\|\tilde{\nabla}g(\theta_n)\|^2 \\
&+ c(1 - p_0)K_0\epsilon_0^2 + c\sqrt{(1 - p_0)K_0}\epsilon_0 \\
<& -\left(\epsilon_0 - cM_0\epsilon_0^2\right)\mathbb{E}\|\tilde{\nabla}g(\theta_n)\|^2 + c\epsilon_0\sqrt{M_0}\,\mathbb{E}\|\tilde{\nabla}g(\theta_n)\| + c(1 - p_0)K_0\epsilon_0^2 \\
&+ c\sqrt{(1 - p_0)K_0}\epsilon_0.
\end{aligned}
$$

Since $\frac{1}{2}\epsilon_0 - cM_0\epsilon_0^2 > 0$, and $\|\tilde{\nabla}g(\theta_n)\| > \max\{4c\sqrt{M_0}, 4c\sqrt{(1 - p_0)K_0}\}$ when $\theta_n \to \infty$, we can get $P(\|\tilde{\nabla}g(\theta_n)\| > \max\{4c\sqrt{M_0}, 4c\sqrt{(1 - p_0)K_0}\}) \to 1$. This implies

$$\mathbb{E}\|\tilde{\nabla}g(\theta_n)\|^2 \geq \left(\mathbb{E}\|\tilde{\nabla}g(\theta_n)\|\right)^2$$

With this, we have

$$\mathbb{E}\left(g(\theta_{n+1})\right) \leq \mathbb{E}\left(g(\theta_1)\right) - \hat{k}_1'\epsilon_0 \sum_{k=1}^n \mathbb{E}\|\tilde{\nabla}g(\theta_n)\|^2 \to -\infty,$$

which is impossible. We thus conclude that $\{\theta_n\}$ can not tend to infinity almost surely, i.e., $P(\theta_n \to \infty) < 1$.

$\square$

## A.4 PROOF OF THEOREM 3.1.

*Proof.* For convenience, we abbreviate $r_{\theta^*} := r$. Then we let

$$l_0 := \min\left\{\left(\frac{\beta_{\theta_*} + 1}{2r(r + 1)G_0^{(r_{\theta^*})}\alpha_{\theta^*}\epsilon_0^r}\right)^{\frac{r+1}{r-1}}, \delta_{\theta^*}\right\},$$

and construct a function

$$R(\theta) = \begin{cases} \|\theta - \theta^*\|^{r+1}, & \text{if } \|\theta - \theta^*\| \leq \max\{1, \delta_{\theta^*}\} \\ \|\theta - \theta^*\|^2, & \text{if } \|\theta - \theta^*\| > \bar{K}_0 \\ \hat{k}(\|\theta - \theta^*\|), & \text{if } \max\{1, \delta_{\theta^*}\} < \|\theta - \theta^*\| \leq \bar{K}_0 \end{cases},$$

where $\hat{k}(\|\theta - \theta^*\|)$ is the smooth connection between $\|\theta - \theta^*\|$ ($\|\theta - \theta^*\| > \bar{K}_0$) and $\|\theta - \theta^*\|^{r+1}$ ($\|\theta - \theta^*\| \leq \max\{1, \delta_{\theta^*}\}$).

Then through choosing feasible $\hat{k}(\theta - \theta^*)$ and $\hat{K}_0$, we can ensure that the Hessian matrix of $R(\theta)$ is bounded in $\mathbb{R}^d$. Let the upper bound of the Hessian matrix be $r(r+1)$, i.e., $x^T H_{\theta\theta} x \leq r(r+1)\|x\|^2$ ($\forall x \in \mathbb{R}^d, \ \theta \in \mathbb{R}^d$).

Next, we construct a set

$$S^{(l_0)} = \left\{\theta \big| 0 \leq \|\theta - \theta^*\| < l_0\right\}.$$

We also define event $A_n^{(l_0)} = \{\theta_n \in S^{(l_0)}\}$ and the characteristic function $I_n^{(l_0)}$. Through the Lagrange's mean value theorem, we obtain

$$I_n^{(l_0)}\left(R(\theta_{n+1}) - R(\theta_n)\right) = I_n^{(l_0)}\nabla R(\theta_{\zeta_n})^T(\theta_{n+1} - \theta_n),$$

where $\theta_{\zeta_n} \in [\theta_{n+1}, \theta_n]$. Note that

$$\nabla R(\theta_{\zeta_n}) = \nabla R(\theta_n) + \nabla R(\theta_{\zeta_n}) - \nabla R(\theta_n),$$

and thus

$$I_n^{(l_0)}\left(R(\theta_{n+1}) - R(\theta_n)\right) \leq -I_n^{(l_0)}\nabla R(\theta_n)^T v_n + I_n^{(l_0)}\|\nabla R(\theta_{\zeta_n}) - \nabla R(\theta_n)\|\|\theta_{n+1} - \theta_n\|.$$

Hence, for any $\theta \in \{\theta | \|\theta - \theta_n\| \leq \max\{1, \delta_{\theta^*}\}\}$ we have

$$\nabla R(\theta) = \nabla\left(\|\theta - \theta^*\|^{r+1}\right) = (r+1)\|\theta - \theta^*\|^{r-1}(\theta - \theta^*).$$

Moreover, if $\|\theta_{\xi_n} - \theta_n\| < \max\{1, \delta_{\theta^*}\}$, we also have

$$\|\nabla R(\theta_{\zeta_n}) - \nabla R(\theta_n)\| \leq r(r+1)\|\theta_{n+1} - \theta_n\|^r,$$

and if $\|\theta_{\xi_n} - \theta_n\| \geq \max\{l_0, 1\}$, we have

$$\begin{aligned}\|\nabla R(\theta_{\zeta_n}) - \nabla R(\theta_n)\| &\leq r(r+1)\|\theta_{\zeta_n} - \theta_n\| \\ &\leq \frac{r(r+1)}{\|\theta_{\zeta_n} - \theta_n\|^{r-1}}\|\theta_{\zeta_n} - \theta_n\|^r \\ &\leq \frac{r(r+1)}{1}\|\theta_{n+1} - \theta_n\|^r.\end{aligned}$$

With this, we have

$$\begin{aligned}\nabla R(\theta_{\zeta_n}) - \nabla R(\theta_n)\| &\leq r(r+1)\|\theta_{n+1} - \theta_n\|^r = r(r+1)\|v_n\|^r, \\ I_n^{(l_0)}\left(R(\theta_{n+1}) - R(\theta_n)\right) &\leq -I_n^{(l_0)}\nabla R(\theta_n)^T v_n + I_n^{(l_0)}r(r+1)\|v_n\|^{r+1}. \\ I_{n+1}^{(l_0)}R(\theta_{n+1}) - I_n^{(l_0)}R(\theta_n) &\leq -I_n^{(l_0)}\nabla R(\theta_n)^T v_n + I_n^{(l_0)}r(r+1)\|v_n\|^{r+1} \\ &\quad - (I_n^{(l_0)} - I_{n+1}^{(l_0)})R(\theta_{n+1}).\end{aligned} \tag{11}$$

Taking expectation of equation 11, we have

$$\begin{aligned}&\mathbb{E}\left(I_n^{(l_0)}\nabla R(\theta_n)^T v_n\right) \\ &= \mathbb{E}\left(I_n^{(l_0)}\mathbb{E}\left(\nabla R(\theta_n)^T v_n \big| \mathcal{F}_n\right)\right) \\ &= \mathbb{E}\left(I_n^{(l_0)}\epsilon_0\mathbb{E}\left(\nabla R(\theta_n)^T\tilde{\nabla}g(\theta_n, \xi_n)\right) + I_n^{(l_0)}\epsilon_0\mathbb{E}\left(\nabla R(\theta_n)^T\sqrt{\min\{g(\theta_n), K_0\}}\tau_n\mathcal{N}_n\big|\mathcal{F}_n\right)\right) \\ &= \epsilon_0\mathbb{E}\left(I_n^{(l_0)}\mathbb{E}\left(\nabla R(\theta_n)^T\tilde{\nabla}g(\theta_n)\right).\end{aligned}$$

Define $\hat{S}$ to be the set of $\theta'$, such that $g(\theta')$ is not smooth. Then with Assumption 2.1, we have $\mathbb{E}_{\theta_n \in \hat{S}}(h(\theta_n)) = 0$, where $h$ is an arbitrary measurable function. Hence, when $\theta_n \in \mathbb{R}^d/\hat{S}$,

$$\begin{aligned}&I_n^{(l_0)}\nabla R(\theta_n)^T\tilde{\nabla}g(\theta_n) \\ &= I_n^{(l_0)}(r+1)\|\theta_n - \theta^*\|^{r-1}(\theta_n - \theta^*)^T\tilde{\nabla}g(\theta_n) \\ &\geq I_n^{(l_0)}(r+1)\|\theta_n - \theta^*\|^{r-1}\alpha_{\theta^*}\|\theta_n - \theta^*\|^{r+1} = I_n^{(l_0)}\alpha_{\theta^*}(r+1)R^{\frac{2r}{r+1}}(\theta_n).\end{aligned}$$

Therefore, we have

$$\mathbb{E}\left(I_n^{(l_0)}\nabla R(\theta_n)^T\tilde{\nabla}g(\theta_n)\right) = \mathbb{E}_{\theta_n\in\mathbb{R}^d/\hat{S}}\left(I_n^{(l_0)}\nabla R(\theta_n)^T\tilde{\nabla}g(\theta_n)\right)$$
$$\geq \mathbb{E}\left(I_n^{(l_0)}\alpha_{\theta^*}(r+1)R^{\frac{2r}{r+1}}(\theta_n)\right),$$

and through Assumption 2.2, we get

$$\mathbb{E}\left(I_n^{(l_0)}r(r+1)\|v_n\|^{r+1}\right)$$
$$= r(r+1)\epsilon_0^{r+1}\mathbb{E}\left(I_n^{(l_0)}\mathbb{E}\left(\|\tilde{\nabla}g(\theta_n,\xi_n)\|^{r+1}|\mathcal{F}_n\right)\right)$$
$$+ r(r+1)\epsilon_0^{r+1}\mathbb{E}\left(I_n^{(l_0)}\mathbb{E}\left(\|\sqrt{\min\{g(\theta_n),K_0\}}\tau_n\mathcal{N}_n\|^{r+1}|\mathcal{F}_n\right)\right)$$
$$+ r(r+1)\epsilon_0^{r+1}\mathbb{E}\left(I_n^{(l_0)}\mathbb{E}\left(\tilde{\nabla}g(\theta_n,\xi_n)^T\sqrt{\min\{g(\theta_n),K_0\}}\tau_n\mathcal{N}_n|\mathcal{F}_n\right)\right)$$
$$= r(r+1)\epsilon_0^{r+1}\mathbb{E}\left(I_n^{(l_0)}\mathbb{E}\left(\|\tilde{\nabla}g(\theta_n,\xi_n)\|^{r+1}|\mathcal{F}_n\right)\right)$$
$$+ r(r+1)\epsilon_0^{r+1}\mathbb{E}\left(I_n^{(l_0)}\mathbb{E}\left(\|\sqrt{\min\{g(\theta_n),K_0\}}\tau_n\mathcal{N}_n\|^{r+1}|\mathcal{F}_n\right)\right)$$
$$\leq r(r+1)(2-p_0)\epsilon_0^{r+1}(\beta_{\theta_*}+1)G_0^{(r_{\theta^*})}\mathbb{E}\left(I_n^{(l_0)}R(\theta_n)\right),$$

where $G_0^{(r_{\theta^*})}$ is defined in Claim 2.1, and results of equation 2 can be seen the situation which $p_0 = 0$. Then,

$$\mathbb{E}\left(I_{n+1}^{(l_0)}R(\theta_{n+1})\right) - \mathbb{E}\left(I_n^{(l_0)}R(\theta_n)\right) \leq -\alpha_{\theta^*}\epsilon_0\mathbb{E}\left(I_n^{(l_0)}R(\theta_n)^{\frac{2r}{r+1}}\right)$$
$$+ r(r+1)(2-p_0)(\beta_{\theta_*}+1)\epsilon_0^{r+1}G_0^{(r_{\theta^*})}\mathbb{E}\left(I_n^{(l_0)}R^{\frac{r+1}{2}}(\theta_n)\right)$$
$$- \mathbb{E}\left((I_n^{(l_0)}-I_{n+1}^{(l_0)})R(\theta_{n+1})\right).$$

Due to $\theta_n \in S^{(l_0)}$, we know

$$R(\theta_n) < \left(\frac{\beta_{\theta_*}+1}{2r(r+1)(2-p_0)G_0^{(r_{\theta^*})}\alpha_{\theta^*}\epsilon_0^r}\right)^{\frac{r+1}{r-1}}.$$

That means

$$\alpha_{\theta^*}\epsilon_0 I_n^{(l_0)}R^{\frac{2r}{r+1}}(\theta_n) > 2r(r+1)(2-p_0)\alpha_{\theta^*}\epsilon_0^{r+1}G_0^{(r_{\theta^*})}I_n^{(l_0)}R(\theta_n).$$

Hence,

$$\mathbb{E}\left(I_{n+1}^{(l_0)}R(\theta_{n+1})\right) - \mathbb{E}\left(I_n^{(l_0)}R(\theta_n)\right)$$
$$\leq -\frac{\alpha_{\theta^*}\epsilon_0}{2}\mathbb{E}\left(I_n^{(l_0)}R^{\frac{2r}{r+1}}(\theta_n)\right) - \mathbb{E}\left((I_n^{(l_0)}-I_{n+1}^{(l_0)})R(\theta_{n+1})\right).$$

For the term $\mathbb{E}\left((I_n^{(l_0)}-I_{n+1}^{(l_0)})R(\theta_{n+1})\right)$, we observe that

$$\mathbb{E}\left((I_n^{(l_0)}-I_{n+1}^{(l_0)})R(\theta_{n+1})\right) = \mathbb{E}\left((I_n^{(l_0)}-I_n^{(l_0)}I_{n+1}^{(l_0)})R(\theta_{n+1})-(I_{n+1}^{(l_0)}-I_n^{(l_0)}I_n^{(l_0)})R(\theta_{n+1})\right),$$
(12)

and

$$(I_n^{(l_0)}-I_n^{(l_0)}I_{n+1}^{(l_0)})g(\theta_{n+1}) \geq l_0(I_n^{(l_0)}-I_n^{(l_0)}I_{n+1}^{(l_0)}),$$
$$(I_{n+1}^{(l_0)}-I_n^{(l_0)}I_n^{(l_0)})g(\theta_{n+1}) \leq l_0(I_{n+1}^{(l_0)}-I_n^{(l_0)}I_n^{(l_0)}).$$

Taking these into equation 12, we obtain

$$\mathbb{E}\left((I_n^{(l_0)}-I_{n+1}^{(l_0)})R(\theta_{n+1})\right) \geq \mathbb{E}\left((I_n^{(l_0)}-I_n^{(l_0)}I_{n+1}^{(l_0)})l_0-(I_{n+1}^{(l_0)}-I_n^{(l_0)}I_n^{(l_0)})l_0\right)$$
$$= l_0\mathbb{E}\left(I_n^{(l_0)}-I_{n+1}^{(l_0)}\right).$$
(13)

Taking equation 13 into equation 12, we have

$$\mathbb{E}\left(I_{n+1}^{(l_0)}R(\theta_{n+1})\right) - \mathbb{E}\left(I_n^{(l_0)}R(\theta_n)\right) \leq -\frac{\alpha_{\theta^*}\epsilon_0}{2}\mathbb{E}\left(I_n^{(l_0)}R^{2r}(\theta_n)\right) - l_0\mathbb{E}\left(I_n^{(l_0)}-I_{n+1}^{(l_0)}\right).$$
(14)

Summing equation 14 over $n$, we have

$$\mathbb{E}\left(I_{n+1}^{(l_0)}R(\theta_{n+1})\right) - \mathbb{E}\left(I_1^{(l_0)}R(\theta_1)\right) \leq -\frac{\alpha_{\theta^*}\epsilon_0}{2}\sum_{k=1}^n \mathbb{E}\left(I_k^{(l_0)}R^{\frac{2r}{r+1}}(\theta_n)\right) - l_0\,\mathbb{E}\left(I_1^{(l_0)} - I_{n+1}^{(l_0)}\right).$$
(15)

Rearranging the equation, we have

$$\sum_{k=1}^n \mathbb{E}\left(I_k^{(l_0)}R^{\frac{2r}{r+1}}(\theta_n)\right) \leq \frac{2(l_0 + g(\theta_1))}{\alpha_{\theta^*}\epsilon_0} < +\infty.$$

Next we construct a subset of $S^{(l_0)}$ as

$$S^{(\delta_0,l_0)} = \left\{\theta \,\middle|\, 0 < \delta_0 \leq \|\theta - \theta^*\| < l_0\right\}.$$

Define event

$$A_n^{(\delta_0,l_0)} = \{\theta_n \in S^{(\delta_0,l_0)}\}$$

and the characteristic function be $I_n^{(\delta_0,l_0)}$. Obviously, we have

$$\sum_{k=1}^n \mathbb{E}\left(I_k^{(\delta_0,l_0)}R^{\frac{2r}{r+1}}(\theta_k)\right) < \sum_{k=1}^n \mathbb{E}\left(I_k^{(l_0)}R^{\frac{2r}{r+1}}(\theta_k)\right) \leq \frac{2(l_0 + g(\theta_1))}{\alpha_{\theta^*}\epsilon_0} < +\infty.$$

Let $r_0 := \inf_{\theta \in S^{(\delta_0,l_0)}} R^{\frac{2r}{r+1}}(\theta) > 0$, we have

$$r_0\sum_{k=1}^n \mathbb{E}\left(I_k^{(\delta_0,l_0)}\right) < \frac{2(l_0 + g(\theta_1))}{\alpha_{\theta^*}\epsilon_0} < +\infty,$$

that is

$$\sum_{k=1}^{+\infty} P\left(\theta_k \in S^{(\delta_0,l_0)}\right) = \sum_{k=1}^{+\infty} \mathbb{E}\left(I_k^{(\delta_0,l_0)}\right) < \frac{2(l_0 + g(\theta_1))}{\alpha_{\theta^*}\epsilon_0 r_0} < +\infty.$$
(16)

Then we can obtain

$$P\left(\{\theta_n\} \in S^{(\delta_0,l_0)}, \text{i.o.}\right) = P\left(\bigcap_{n=1}^{+\infty}\bigcup_{k=n}^{+\infty}\left(\theta_k \in S^{(\delta_0,l_0)}\right)\right)$$
(17)

$$= \lim_{n\to+\infty} P\left(\bigcup_{k=n}^{+\infty}\left(\theta_k \in S^{(\delta_0,l_0)}\right)\right)$$
(18)

$$\leq \lim_{n\to+\infty}\sum_{k=n}^{+\infty} P\left(\theta_k \in S^{(\delta_0,l_0)}\right) = 0.$$
(19)

Note that equation 17 means the set $S^{(\delta_0,l_0)}$ has no limit point of $\{\theta_n\}$ almost surely. Then if we use the SGD update rule equation 5 Since the noise is Gaussian, any $\theta \in \mathbb{R}^d/J^*$ and for any $k > 0$, there is $P(\theta_{n+k} \in S^{(\delta_0,l_0)}|\theta_n = \theta) = \hat{\delta}_0 > 0$. If we use SGD update rule equation 2, for any max positive invariant set $D/J^*$, we know that there must exist a boundary set $\partial D$. Moreover, $\forall \theta' \in \partial D$, if $\theta' \in \mathbb{R}^d/D$, then for any mini-batch $C_i$, we have $\tilde{\nabla}g_{C_i}(\theta') = 0$. Otherwise we can find a sequence $\{\theta'' \to \theta', /\theta'' \in D\}$, making the trajectories started from $\theta''$ close to the trajectory started from $\theta'$. It forms a contradiction. Then due to $J^{**} = J^*$, we know $\theta' \in J^*$. That means $\overline{D} \cap J^* \neq \emptyset$. If $\theta' \in D$, we can conclude all trajectories started from $\theta'$ are a subset of $\partial D$. On the other hand, we can conclude $\partial g$ is a close set. Through $Heine\check{\ }Borel\ theorem$, it exists a finite open cover $\bigcup_{n=1}^M O_n \supset \partial D$, and every $O_n$ holding an arbitrary small diameter. We let $\theta' \in O_1$. Then we assign $T_n$ as the lone time interval of one trajectory started from $\theta'$ and back to $T_n$. If $T_n \to +\infty$, that means this trajectory must stay a infinity time in some $O_k$, that means exists a global optimum in $O_k$. Naturally, the trajectory will converge to this global optimum. If $T_n$ is bounded, that means the trajectory will enter into $O_1$ infinite times. Due to a mass of different mini-batch and the enough small diameter and $f(\theta) := P(\theta_{n+k} \in \mathbb{R}^d/D|\theta_n = \theta) = \hat{\delta}_0 > 0$ is a continuous function, We get $P(\theta_{n+k} \in \mathbb{R}^d/D|\theta_n \in O_1) = \hat{\delta}_0 > 0$, it is contradiction about $D$ is a

positive invariant set. That means for any $\theta \in \mathbb{R}/J^*$, either trajectories started from it will converge to some global optimum, either it has a positive probability to make sure it transfers to $S^{(\delta_0, l_0)}$ after $k$ steps. Then for any bounded set $\hat{S}_0$ which has no intersection with $J^*$, we first get rid of those points which will converge to $J^*$. We know that $f(\theta) := P(\theta_{n+k} \in S^{(\delta_0, l_0)} | \theta_n = \theta) = \hat{\delta}_0 > 0$ is a continuous function. Then we can get for any bounded closed set $\hat{S}_0$ which satisfied $\hat{S}_0 \cap J^* = \emptyset$, there is $\min_{\theta \in \hat{S}_0} P(\theta_{n+k} \in S^{(\delta_0, l_0)} | \theta_n = \theta) = \hat{\delta}_1 > 0$. Then we aim to prove there is no limit point in $\hat{S}_0$ almost surely by contradiction. We assume

$$\sum_{n=1}^{+\infty} P(\theta_n \in \hat{S}_0) = +\infty.$$

Then,

$$
\begin{aligned}
\sum_{n=k+1}^{+\infty} P(\theta_n \in S^{(\delta_0, l_0)}) &= \sum_{n=k+1}^{+\infty} \int_{S^{(\delta_0, l_0)}} P_n(d\theta) \\
&= \sum_{n=k+1}^{+\infty} \int_{S^{\mathbb{R}^d}} P(\theta_{n+k} \in S^{(\delta_0, l_0)} | \theta_n = \theta) P_{n-k}(d\theta) \\
&\geq \sum_{n=k+1}^{+\infty} \int_{\hat{S}_0} P(\theta_{n+k} \in S^{(\delta_0, l_0)} | \theta_n = \theta) P_{n-k}(d\theta) \\
&\geq \hat{\delta}_1 \sum_{n=k+1}^{+\infty} \int_{\hat{S}_0} P_{n-k}(d\theta) = \hat{\delta}_1 \sum_{n=1}^{+\infty} P(\theta_n \in \hat{S}_0) \\
&= +\infty.
\end{aligned}
$$

Note that this is in contradiction with equation 16 and thus $\sum_{n=1}^{+\infty} P(\theta_n \in \hat{S}_0) < +\infty$. Then,

$$
\begin{aligned}
P(\{\theta_n\} \in \hat{S}_0, \text{ i.o.}) &= P\left(\bigcap_{n=1}^{+\infty} \bigcup_{k=n}^{+\infty} (\theta_k \in \hat{S}_0)\right) \\
&= \lim_{n \to +\infty} P\left(\bigcup_{k=n}^{+\infty} (\theta_k \in \hat{S}_0)\right) \qquad (20) \\
&\leq \lim_{n \to +\infty} \sum_{k=n}^{+\infty} P(\theta_k \in \hat{S}_0) = 0.
\end{aligned}
$$

Combining equation 20 with equation 16, we can see that for any bounded set which does not include $J^* = \{\theta | g(\theta) = 0\}$ has no limit point almost surely. This implies $\theta_n \to J^*$ or $\theta_n \to \infty$. Since $\{\{\theta_n\}$ is convergence$\}$ is a tail event. Then by the zero-one law, we know $P(\{\theta_n\}$ is convergence$) = 0$ or $1$. That means $\{\theta_n\}$ either converges to $J^*$ almost surely, or diverges to infinity almost surely. Through Lemma A.4, we know $P(\theta_n \to \infty) < 1$, thus $\{\theta_n\}$ can only converge to $J^*$ almost surely. $\qquad \square$

### A.5 PROOF OF THEOREM 3.2

*Proof.* We define $R(\theta) = \|\theta - \theta^*\|^2$, and a set

$$S^{(l_0)} = \{\theta | 0 \leq \|\theta - \theta^*\| < l_0 := \delta_{\theta^*}\}.$$

We also define an event $A_n^{(l_0)} = \{\theta_n \in S^{(l_0)}\}$ and the characteristic function $I_n^{(l_0)}$. By Lagrange's mean value theorem, we have

$$I_n^{(l_0)}(R(\theta_{n+1}) - R(\theta_n)) = I_n^{(l_0)} \nabla R(\theta_{\zeta_n})^T (\theta_{n+1} - \theta_n),$$

where $\theta_{\zeta_n} \in [\theta_{n+1}, \theta_n]$.

Note that $\nabla R(\theta_{\zeta_n}) = \nabla R(\theta_n) + \nabla R(\theta_{\zeta_n}) - \nabla R(\theta_n)$, we have

$$I_n^{(l_0)}(R(\theta_{n+1}) - R(\theta_n)) \leq -I_n^{(l_0)} \nabla R(\theta_n)^T v_n + I_n^{(l_0)} \|\nabla R(\theta_{\zeta_n}) - \nabla R(\theta_n)\| \|\theta_{n+1} - \theta_n\|.$$

Moreover, we also have

$$\|\nabla R(\theta_{\zeta_n}) - \nabla R(\theta_n)\| \le 2\|\theta_{n+1} - \theta_n\| = 2\|v_n\|$$
$$I_n^{(l_0)}\big(R(\theta_{n+1}) - R(\theta_n)\big) \le -I_n^{(l_0)}\nabla R(\theta_n)^T v_n + I_n^{(l_0)}2\|v_n\|^2$$
$$I_n^{(l_0)}\big(R(\theta_{n+1}) - R(\theta_n)\big) \le -I_n^{(l_0)}\nabla R(\theta_n)^T v_n + I_n^{(l_0)}2\|v_n\|^2$$
$$I_{n+1}^{(l_0)}R(\theta_{n+1}) - I_n^{(l_0)}R(\theta_n) \le -I_n^{(l_0)}\nabla R(\theta_n)^T v_n + I_n^{(l_0)}2\|v_n\|^2 \tag{21}$$
$$- (I_n^{(l_0)} - I_{n+1}^{(l_0)})R(\theta_{n+1}). \tag{22}$$

Taking expectation of equation 21, we have

$$\mathbb{E}\left(I_n^{(l_0)}\nabla R(\theta_n)^T v_n\right)$$
$$= \mathbb{E}\left(I_n^{(l_0)}\mathbb{E}\left(\nabla R(\theta_n)^T v_n | \mathcal{F}_n\right)\right)$$
$$= \mathbb{E}\left(I_n^{(l_0)}\epsilon_0\,\mathbb{E}\left(\nabla R(\theta_n)^T \tilde{\nabla}g(\theta_n, \xi_n)\right) + I_n^{(l_0)}\epsilon_0\,\mathbb{E}\left(\nabla R(\theta_n)^T \sqrt{\min\{g(\theta_n), K_0\}}\tau_n \mathcal{N}_n\big|\mathcal{F}_n\right)\right)$$
$$= \epsilon_0\,\mathbb{E}\left(I_n^{(l_0)}\epsilon_0\,\mathbb{E}\left(\nabla R(\theta_n)^T \tilde{\nabla}g(\theta_n)\right)\right).$$

We define $\hat{S} = \{\theta' | \tilde{\nabla}g(\theta)$ is not continue at $\theta'\}$. Then through Assumption 2.1, and note that $\mathbb{E}_{\theta_n \in \hat{S}}(h(\theta_n)) = 0$, where $h$ is an arbitrary measurable function, we have that the following when $\theta_n \in \mathbb{R}^d/\hat{S}$.

$$I_n^{(l_0)}\nabla R(\theta_n)^T \tilde{\nabla}g(\theta_n) = 2I_n^{(l_0)}(\theta_n - \theta^*)^T \tilde{\nabla}g(\theta_n) \ge 2I_n^{(l_0)}\alpha_{\theta^*}\|\theta_n - \theta^*\|^2$$
$$\ge 2I_n^{(l_0)}\alpha_{\theta^*}\|\theta_n - \theta^*\|^2 = 2I_n^{(l_0)}\alpha_{\theta^*}R(\theta_n).$$

Therefore, we have

$$\mathbb{E}\left(I_n^{(l_0)}\nabla R(\theta_n)^T \tilde{\nabla}g(\theta_n)\right) = \mathbb{E}_{\theta_n \in \mathbb{R}^d/\hat{S}}\left(I_n^{(l_0)}\nabla R(\theta_n)^T \tilde{\nabla}g(\theta_n)\right)$$
$$\ge 2\,\mathbb{E}\left(I_n^{(l_0)}\alpha_{\theta^*}R(\theta_n)\right).$$

and through Assumption 2.2, we get

$$\mathbb{E}\left(I_n^{(l_0)}2\|v_n\|^2\right) = 2\epsilon_0^2\,\mathbb{E}\left(I_n^{(l_0)}\mathbb{E}\left(\|\tilde{\nabla}g(\theta_n, \xi_n)\|^2 | \mathcal{F}_n\right)\right)$$
$$+ 2\epsilon_0^2\,\mathbb{E}\left(I_n^{(l_0)}\mathbb{E}\left(\|\sqrt{\min\{g(\theta_n), K_0\}}\tau_n \mathcal{N}_n\|^2 | \mathcal{F}_n\right)\right)$$
$$+ 4\epsilon_0^2\,\mathbb{E}\left(I_n^{(l_0)}\mathbb{E}\left(\tilde{\nabla}g(\theta_n, \xi_n)^T \sqrt{\min\{g(\theta_n), K_0\}}\tau_n \mathcal{N}_n | \mathcal{F}_n\right)\right)$$
$$= 2\epsilon_0^2\,\mathbb{E}\left(I_n^{(l_0)}\mathbb{E}\left(\|\tilde{\nabla}g(\theta_n, \xi_n)\|^2 | \mathcal{F}_n\right)\right)$$
$$+ 2\epsilon_0^2\,\mathbb{E}\left(I_n^{(l_0)}\mathbb{E}\left(\|\sqrt{\min\{g(\theta_n), K_0\}}\tau_n \mathcal{N}_n\|^2 | \mathcal{F}_n\right)\right)$$
$$\le 2(2 - p_0)\epsilon_0^2\beta_{\theta^*}^2\,\mathbb{E}\left(I_n^{(l_0)}R(\theta_n)\right),$$

where the situation of equation 2 can be seen as $p_0 = 0$. Then we have

$$\mathbb{E}\left(I_{n+1}^{(l_0)}R(\theta_{n+1})\right) - \mathbb{E}\left(I_n^{(l_0)}R(\theta_n)\right)$$
$$\le -c_{\theta^*}\epsilon_0\,\mathbb{E}\left(I_n^{(l_0)}R(\theta_n)\right) + 2(2 - p_0)\epsilon_0^2\beta_{\theta^*}^2\,\mathbb{E}\left(I_n^{(l_0)}R(\theta_n)\right) - \mathbb{E}\left((I_n^{(l_0)} - I_{n+1}^{(l_0)})R(\theta_{n+1})\right),$$

and

$$\mathbb{E}\left(I_{n+1}^{(l_0)}R(\theta_{n+1})\right) - \mathbb{E}\left(I_n^{(l_0)}R(\theta_n)\right)$$
$$\le -\left(\alpha_{\theta^*}\epsilon_0 - 2(2 - p_0)\epsilon_0^2\beta_{\theta^*}^2\right)\mathbb{E}\left(I_n^{(l_0)}R(\theta_n)\right) - \mathbb{E}\left((I_n^{(l_0)} - I_{n+1}^{(l_0)})R(\theta_{n+1})\right).$$

For the term $\mathbb{E}\left((I_n^{(l_0)} - I_{n+1}^{(l_0)})R(\theta_{n+1})\right)$, we first observe that

$$\mathbb{E}\left((I_n^{(l_0)} - I_{n+1}^{(l_0)})R(\theta_{n+1})\right) = \mathbb{E}\left((I_n^{(l_0)} - I_n^{(l_0)}I_{n+1}^{(l_0)})R(\theta_{n+1}) - (I_{n+1}^{(l_0)} - I_n^{(l_0)}I_n^{(l_0)})R(\theta_{n+1})\right), \tag{23}$$

and

$$(I_n^{(l_0)} - I_n^{(l_0)} I_{n+1}^{(l_0)}) g(\theta_{n+1}) \geq l_0 (I_n^{(l_0)} - I_n^{(l_0)} I_{n+1}^{(l_0)}),$$
$$(I_{n+1}^{(l_0)} - I_n^{(l_0)} I_n^{(l_0)}) g(\theta_{n+1}) \leq l_0 (I_{n+1}^{(l_0)} - I_n^{(l_0)} I_n^{(l_0)}).$$

Taking these into equation 23, we have

$$\mathbb{E}\left((I_n^{(l_0)} - I_{n+1}^{(l_0)}) R(\theta_{n+1})\right) \geq \mathbb{E}\left((I_n^{(l_0)} - I_n^{(l_0)} I_{n+1}^{(l_0)}) l_0 - (I_{n+1}^{(l_0)} - I_n^{(l_0)} I_n^{(l_0)}) l_0\right)$$
$$= l_0 \,\mathbb{E}\left(I_n^{(l_0)} - I_{n+1}^{(l_0)}\right). \tag{24}$$

Substituting equation 24 into equation 23, we get

$$\mathbb{E}\left(I_{n+1}^{(l_0)} R(\theta_{n+1})\right) - \mathbb{E}\left(I_n^{(l_0)} R(\theta_n)\right)$$
$$\leq -\left(\alpha_{\theta^*} \epsilon_0 - 2(2 - p_0)\epsilon_0^2 \beta_{\theta^*}^2\right) \mathbb{E}\left(I_n^{(l_0)} R(\theta_n)\right) - l_0 \,\mathbb{E}\left(I_n^{(l_0)} - I_{n+1}^{(l_0)}\right). \tag{25}$$

Summing equation 25 over $n$, we have

$$\mathbb{E}\left(I_{n+1}^{(l_0)} R(\theta_{n+1})\right) - \mathbb{E}\left(I_1^{(l_0)} R(\theta_1)\right)$$
$$\leq -\left(\alpha_{\theta^*} \epsilon_0 - 2(2 - p_0)\epsilon_0^2 \beta_{\theta^*}^2\right) \sum_{k=1}^{n} \mathbb{E}\left(I_k^{(l_0)} R(\theta_n)\right) - l_0 \,\mathbb{E}\left(I_1^{(l_0)} - I_{n+1}^{(l_0)}\right). \tag{26}$$

As $\epsilon_0 < \alpha_{\theta^*}/2(2 - p_0)\beta_{\theta^*}^2$, we have

$$\sum_{k=1}^{n} \mathbb{E}\left(I_k^{(l_0)} R(\theta_n)\right) \leq \frac{l_0 + g(\theta_1)}{\alpha_{\theta^*} \epsilon_0 - 2(2 - p_0)\epsilon_0^2 \beta_{\theta^*}^2} < +\infty.$$

Next, we construct a subset of $S^{(l_0)}$ as

$$S^{(\delta_0, l_0)} = \{\theta | 0 < \delta \leq \|\theta - \theta^*\| < l_0\}.$$

We also define $A_n^{(\delta_0, l_0)} = \{\theta_n \in S^{(\delta_0, l_0)}\}$ and the characteristic function be $I_n^{(\delta_0, l_0)}$. Notice that, we have

$$\sum_{k=1}^{n} \mathbb{E}\left(I_k^{(\delta_0, l_0)} R(\theta_k)\right) < \sum_{k=1}^{n} \mathbb{E}\left(I_k^{(l_0)} R(\theta_k)\right) \leq \frac{l_0 + g(\theta_1)}{\alpha_{\theta^*} \epsilon_0 - 2(2 - p_0)\epsilon_0^2 \beta_{\theta^*}^2} < +\infty.$$

Denote $r_0 := \inf_{\theta \in S^{(\delta_0, l_0)}} R(\theta) > 0$, then

$$r_0 \sum_{k=1}^{n} \mathbb{E}\left(I_k^{(\delta_0, l_0)}\right) < \frac{l_0 + g(\theta_1)}{\alpha_{\theta^*} \epsilon_0 - 2(2 - p_0)\epsilon_0^2 \beta_{\theta^*}^2} < +\infty,$$

that is

$$\sum_{k=1}^{+\infty} P\left(\theta_k \in S^{(\delta_0, l_0)}\right) = \sum_{k=1}^{+\infty} \mathbb{E}\left(I_k^{(\delta_0, l_0)}\right) < \frac{l_0 + g(\theta_1)}{\alpha_{\theta^*} \epsilon_0 - 2(2 - p_0)\epsilon_0^2 \beta_{\theta^*}^2} < +\infty. \tag{27}$$

With this, we have

$$P\left(\{\theta_n\} \in S^{(\delta_0, l_0)}, \text{ i.o.}\right) = P\left(\bigcap_{n=1}^{+\infty} \bigcup_{k=n}^{+\infty} \left(\theta_k \in S^{(\delta_0, l_0)}\right)\right)$$
$$= \lim_{n \to +\infty} P\left(\bigcup_{k=n}^{+\infty} \left(\theta_k \in S^{(\delta_0, l_0)}\right)\right)$$
$$\leq \lim_{n \to +\infty} \sum_{k=n}^{+\infty} P\left(\theta_k \in S^{(\delta_0, l_0)}\right)$$
$$= 0. \tag{28}$$

We remark thatequation 28 implies the set $S^{(\delta_0, l_0)}$ has no limit point of $\{\theta_n\}$ almost surely. Then if we use the SGD update rule equation 5, as the noise is Gaussian, for any $\theta \in \mathbb{R}^d / J^*$ and any $k > 0$, there is $P(\theta_{n+k} \in S^{(\delta_0, l_0)} | \theta_n = \theta) = \hat{\delta}_0 > 0$.

If we use SGD update rule equation 2, for any max positive invariant set $D/J^*$, we know that there must exist a boundary set $\partial D$. Moreover, $\forall \theta' \in \partial D$, if $\theta' \in \mathbb{R}^d/D$, then for any mini-batch $C_i$, we have $\tilde{\nabla} g_{C_i}(\theta') = 0$. Otherwise we can find a sequence $\{\theta'' \to \theta', /\theta'' \in D\}$, making the trajectories started from $\theta''$ close to the trajectory started from $\theta'$. It forms a contradiction. Then due to $J^{**} = J^*$, we know $\theta' \in J^*$. That means $\overline{D} \cap J^* \neq \emptyset$. If $\theta' \in D$, we can conclude all trajectories started from $\theta'$ are a subset of $\partial D$. On the other hand, we can conclude $\partial g$ is a close set. Through $Heine\breve{}Borel\ theorem$, it exists a finite open cover $\bigcup_{n=1}^{M} O_n \supset \partial D$, and every $O_n$ holding an arbitrary small diameter. We let $\theta' \in O_1$. Then we assign $T_n$ as the lone time interval of one trajectory started from $\theta'$ and back to $T_n$. If $T_n \to +\infty$, that means this trajectory must stay a infinity time in some $O_k$, that means exists a global optimum in $O_k$. Naturally, the trajectory will converge to this global optimum. If $T_n$ is bounded, that means the trajectory will enter into $O_1$ infinite times. Due to a mass of different mini-batch and the enough small diameter and $f(\theta) := P(\theta_{n+k} \in \mathbb{R}^d/D | \theta_n = \theta) = \hat{\delta}_0 > 0$ is a continuous function, We get $P(\theta_{n+k} \in \mathbb{R}^d/D | \theta_n \in O_1) = \hat{\delta}_0 > 0$, it is contradiction about $D$ is a positive invariant set. That means for any $\theta \in \mathbb{R}/J^*$, either trajectories started from it will converge to some global optimum, either it has a positive probability to make sure it transfers to $S^{(\delta_0, l_0)}$ after $k$ steps. Then for any bounded set $\hat{S}_0$ which has no intersection with $J^*$, we first get rid of those points which will converge to $J^*$. We know that $f(\theta) := P(\theta_{n+k} \in S^{(\delta_0, l_0)} | \theta_n = \theta) = \hat{\delta}_0 > 0$ is a continuous function. Then we can get for any bounded closed set $\hat{S}_0$ which satisfied $\hat{S}_0 \cap J^* = \emptyset$, there is $\min_{\theta \in \hat{S}_0} P(\theta_{n+k} \in S^{(\delta_0, l_0)} | \theta_n = \theta) = \hat{\delta}_1 > 0$. Then we aim to prove there is no limit point in $\hat{S}_0$ almost surely by contradiction. We assume

$$\sum_{n=1}^{+\infty} P(\theta_n \in \hat{S}_0) = +\infty.$$

Then we can get

$$\begin{aligned}
\sum_{n=k+1}^{+\infty} P(\theta_n \in S^{(\delta_0, l_0)}) &= \sum_{n=k+1}^{+\infty} \int_{S^{(\delta_0, l_0)}} P_n(d\theta) \\
&= \sum_{n=k+1}^{+\infty} \int_{S^{\mathbb{R}^d}} P(\theta_{n+k} \in S^{(\delta_0, l_0)} | \theta_n = \theta) P_{n-k}(d\theta) \\
&\geq \sum_{n=k+1}^{+\infty} \int_{\hat{S}_0} P(\theta_{n+k} \in S^{(\delta_0, l_0)} | \theta_n = \theta) P_{n-k}(d\theta) \\
&\geq \hat{\delta}_1 \sum_{n=k+1}^{+\infty} \int_{\hat{S}_0} P_{n-k}(d\theta) = \hat{\delta}_1 \sum_{n=1}^{+\infty} P(\theta_n \in \hat{S}_0) \\
&= +\infty.
\end{aligned}$$

This is contradiction with equation 27, which implies

$$\sum_{n=1}^{+\infty} P(\theta_n \in \hat{S}_0) < +\infty.$$

Hence, we can obtain

$$
\begin{aligned}
P\big(\{\theta_n\} \in \hat{S}_0,\ \text{i.o.}\big) &= P\Big(\bigcap_{n=1}^{+\infty} \bigcup_{k=n}^{+\infty} \big(\theta_k \in \hat{S}_0\big)\Big) \\
&= \lim_{n \to +\infty} P\Big(\bigcup_{k=n}^{+\infty} \big(\theta_k \in \hat{S}_0\big)\Big) \\
&\le \lim_{n \to +\infty} \sum_{k=n}^{+\infty} P\big(\theta_k \in \hat{S}_0\big) \\
&= 0 \,.
\end{aligned}
\tag{29}
$$

Combining equation 29 with equation 27, for any bounded set which does not include $J^* = \{\theta | g(\theta) = 0\}$, we can say that it has no limit point almost surely. That means $\theta_n \to J^*$ or $\theta_n \to \infty$ almost surely. We know the event $\{\theta_n$ is convergence$\}$ is a tail event. By zero-one law, we have $P(\{\theta_n\}$ is convergence$) = 0\ or\ 1$. That means $\{g(\theta_n)\}$ either converges to $J^*$ almost surely, or diverges to infinity almost surely. Through Lemma A.4, we know $P(\theta_n \to \infty) < 1$. That proves $\{\theta_n\}$ can only converge to $J^*$ almost surely. □

## A.6 PROOF OF THEOREM 3.3

First we construct a function $R(\theta) = \|\theta - \theta^*\|^2$. We can get that

$$
\begin{aligned}
R(\theta_{n+1}) - R(\theta_n) &= \|\theta_{n+1} - \theta^*\|^2 - \|\theta_n - \theta^*\|^2 = (\theta_{n+1} - \theta_n)^T(\theta_{n+1} + \theta_n - 2\theta^*) \\
&= 2(\theta_n - \theta^*)^T(\theta_{n+1} - \theta_n) + \|\theta_{n+1} - \theta_n\|^2 = -2(\theta_n - \theta^*)^T v_n + \|v_n\|^2 \\
&= -2(\theta_n - \theta^*)^T\big(\epsilon_0 \tilde{\nabla} g(\theta_n, \xi_n) + \epsilon_0 \sqrt{\min\{g(\theta_n), K_0\}} \tau_n \mathcal{N}_n\big) \\
&\quad + \big\|\epsilon_0 \tilde{\nabla} g(\theta_n, \xi_n) + \epsilon_0 \sqrt{\min\{g(\theta_n), K_0\}} \tau_n \mathcal{N}_n\big\|^2 \,.
\end{aligned}
\tag{30}
$$

For the term $2(\theta_n - \theta^*)^T\big(\epsilon_0 \tilde{\nabla} g(\theta_n, \xi_n) + \epsilon_0 \sqrt{\min\{g(\theta_n), K_0\}} \tau_n \mathcal{N}_n\big)$, we use the following transformation:

$$
\begin{aligned}
&2(\theta_n - \theta^*)^T\big(\epsilon_0 \tilde{\nabla} g(\theta_n, \xi_n) + \epsilon_0 \sqrt{\min\{g(\theta_n), K_0\}} \tau_n \mathcal{N}_n\big) + 2\epsilon_0 \\
&= 2\epsilon_0(\theta_n - \theta^*)^T \tilde{\nabla} g(\theta_n) + 2\epsilon_0(\theta_n - \theta^*)^T\big(\tilde{\nabla} g(\theta_n, \xi_n) - \tilde{\nabla} g(\theta_n)\big) \\
&\quad + 2\epsilon_0 \sqrt{\min\{g(\theta_n), K_0\}} \tau_n (\theta_n - \theta^*)^T \mathcal{N}_n \,.
\end{aligned}
\tag{31}
$$

For the term $\big\|\epsilon_0 \tilde{\nabla} g(\theta_n, \xi_n) + \epsilon_0 \sqrt{\min\{g(\theta_n), K_0\}} \tau_n \mathcal{N}_n\big\|^2$, we can obtain

$$
\begin{aligned}
&\big\|\epsilon_0 \tilde{\nabla} g(\theta_n, \xi_n) + \epsilon_0 \sqrt{\min\{g(\theta_n), K_0\}} \tau_n \mathcal{N}_n\big\|^2 \\
&= \epsilon_0^2\big\|\tilde{\nabla} g(\theta_n, \xi_n)\big\|^2 + 2\epsilon_0^2 \tau_n \sqrt{\min\{g(\theta_n), K_0\}} \tilde{\nabla} g(\theta_n, \xi_n)^T \mathcal{N}_n + \epsilon_0^2 \tau_n^2 \mathcal{N}_n^2 \min\{g(\theta_n), K_0\} \\
&= \epsilon_0^2 \mathbb{E}\Big(\big\|\tilde{\nabla} g(\theta_n, \xi_n)\big\|^2 \big| \mathcal{F}_n\Big) + \epsilon_0^2 p_0 \min\{g(\theta_n), K_0\} + \epsilon_0^2\big\|\tilde{\nabla} g(\theta_n, \xi_n)\big\|^2 \\
&\quad - \epsilon_0^2 \mathbb{E}\Big(\big\|\tilde{\nabla} g(\theta_n, \xi_n)\big\|^2 \big| \mathcal{F}_n\Big) + \epsilon_0^2 \tau_n^2 \mathcal{N}_n^2 \min\{g(\theta_n), K_0\} - \epsilon_0^2 p_0 \min\{g(\theta_n), K_0\} \\
&\quad + 2\epsilon_0^2 \tau_n \sqrt{\min\{g(\theta_n), K_0\}} \tilde{\nabla} g(\theta_n, \xi_n)^T \mathcal{N}_n \\
&\ge \epsilon_0^2\big\|\tilde{\nabla} g(\theta_n)\big\|^2 + \epsilon_0^2\big\|\tilde{\nabla} g(\theta_n, \xi_n)\big\|^2 - \epsilon_0^2 \mathbb{E}\Big(\big\|\tilde{\nabla} g(\theta_n, \xi_n)\big\|^2 \big| \mathcal{F}_n\Big) \\
&\quad + 2\epsilon_0^2 \tau_n \sqrt{\min\{g(\theta_n), K_0\}} \tilde{\nabla} g(\theta_n, \xi_n)^T \mathcal{N}_n \,.
\end{aligned}
\tag{32}
$$

Then we construct a set

$$
S^{(\hat{l}_0)} = \{\theta | \|\theta - \theta^*\| < l_0 := \delta_{\theta^*}\}/\{\theta^*\} \,.
$$

We also define event $A_{i,n} = \{\theta_{n_0} \in S^{(\hat{l}_0)},\ n_0 \in [i, n]\}$, and its characteristic function as $I_{i,n}$. We substitute equation 32 and equation 31 into equation 30, and multiple $I_{i,n}$, getting

$$
I_{i,n}\big(R(\theta_{n+1}) - R(\theta_n)\big) \ge \big(2(2 - p_0)\epsilon_0^2 \alpha_{\theta^*}^2 - \epsilon_0 \beta_{\theta^*}\big) I_{i,n} R(\theta_n) + I_{i,n} \zeta_n,
$$

where

$$\zeta_n := 2\epsilon_0(\theta_n - \theta^*)^T\big(\tilde{\nabla}g(\theta_n, \xi_n) - \tilde{\nabla}g(\theta_n)\big) + 2\epsilon_0\sqrt{\min\{g(\theta_n), K_0\}}\tau_n(\theta_n - \theta^*)^T\mathcal{N}_n$$
$$+ \epsilon_0^2\big\|\tilde{\nabla}g(\theta_n, \xi_n)\big\|^2 - \epsilon_0^2\mathbb{E}\Big(\big\|\tilde{\nabla}g(\theta_n, \xi_n)\big\|^2\big|\mathcal{F}_n\Big) + \epsilon_0^2\tau_n^2\mathcal{N}_n^2\min\{g(\theta_n), K_0\} \qquad (33)$$
$$- \epsilon_0^2 p_0\min\{g(\theta_n), K_0\}$$

is a Martingale difference. Denote $\hat{p}_0 := \big(R(\theta_{n+1}) - R(\theta_n)\big) \geq \big(2(2 - p_0)\epsilon_0^2\alpha_{\theta^*}^2 - \epsilon_0\beta_{\theta^*}\big)$, we have

$$I_{i,n+1}R(\theta_{n+1}) - I_{i,n}R(\theta_n) \geq \hat{p}_0 I_{i,n}R(\theta_n) + I_{i,n}\hat{\zeta}_n - R(\theta_{n+1})(I_{i,n} - I_{i,n+1})\,.$$

Then,

$$\mathbb{E}\big(I_{i,n+1}R(\theta_{n+1})\big) - \mathbb{E}\big(I_{i,n}R(\theta_n)\big) \geq \hat{p}_0\,\mathbb{E}\big(I_{i,n}R(\theta_n)\big) - \mathbb{E}\big(R(\theta_{n+1})(I_{i,n} - I_{i,n+1})\big)\,,$$

which implies

$$\mathbb{E}\big(I_{i,n+1}R(\theta_{n+1})\big) \geq \left(1 + \hat{p}_0 - \frac{\mathbb{E}\big(R(\theta_{n+1})(I_{i,n} - I_{i,n+1})\big)}{\mathbb{E}\big(I_{i,n}R(\theta_n)\big)}\right)\mathbb{E}\big(I_{i,n}R(\theta_n)\big)\,.$$

Assuming

$$\limsup_{n\to+\infty}\frac{\mathbb{E}\big(R(\theta_{n+1})(I_{i,n} - I_{i,n+1})\big)}{\mathbb{E}\big(I_{i,n}R(\theta_n)\big)} < \hat{p}_0\,,$$

we have

$$\mathbb{E}\big(I_{i,n+1}R(\theta_{n+1})\big) \to +\infty\,.$$

Note that this contradicted the $\mathbb{E}\big(I_{i,n+1}R(\theta_{n+1})\big) \leq \hat{l}_0$. Hence,

$$\limsup_{n\to+\infty}\frac{\mathbb{E}\big(R(\theta_{n+1})(I_{i,n} - I_{i,n+1})\big)}{\mathbb{E}\big(I_{i,n}R(\theta_n)\big)} \geq \hat{p}_0. \qquad (34)$$

Define an event $A_{i,+\infty} := \{\theta_{n_0} \in S^{(\hat{l}_0)},\ n_0 \geq i\}$, and its characteristic function as $I_{i,+\infty}$. We next prove $P\Big(\lim_{n\to+\infty}I_{i,+\infty}R(\theta_n) = 0\Big) = 0$.

We assume $P\Big(\lim_{n\to+\infty}I_{i,+\infty}R(\theta_n) = 0\Big) = 1$, and we can get $P\Big(\lim_{n\to+\infty}I_{i,n}R(\theta_n) = 0\Big) = 1$. That means for any $\epsilon_0' > 0$, $P\Big(I_{i,n}R(\theta_n) > \epsilon_0'\Big) \to 0$, concluding $P\Big(I_{i,n}R(\theta_n) \leq \epsilon_0'\Big) \to 1$. Then we get

$$\limsup_{n\to+\infty}\frac{\mathbb{E}\big(R(\theta_{n+1})(I_{i,n} - I_{i,n+1})\big)}{\mathbb{E}\big(I_{i,n}R(\theta_n)\big)} = \limsup_{n\to+\infty}\frac{\int\mathbb{E}(R(\theta_{n+1}) > \hat{l}_0|\theta = \theta)P_{i,n}(d\theta)}{\int_{R(\theta)\epsilon_0'}R(\theta)P_{i,n}(d\theta) + \int_{R(\theta)>\epsilon_0'}R(\theta)P_{i,n}(d\theta)}$$
$$= \limsup_{n\to+\infty}\frac{\int_{R(\theta)\leq\epsilon_0'}\mathbb{E}(R(\theta_{n+1}) > \hat{l}_0|\theta = \theta)P_{i,n}(d\theta)}{\int_{R(\theta)\leq\epsilon_0'}R(\theta)P_{i,n}(d\theta)}$$
$$< \frac{\hat{p}_0}{2}\,.$$

Note that this contradicted equation 34, which implies $P\Big(\lim_{n\to+\infty}I_{i,+\infty}R(\theta_n) = 0\Big) = 0$. Through inspecting the event $\{\theta_n \to \theta^*\}$, we can get

$$\{\theta_n \to \theta^*\} \subset \left\{\bigcup_{i=1}^{+\infty}A_{i,+\infty}\right\}\,.$$

That means

$$P\big(\theta_n \to \theta^*\big) = P\bigg( \{\theta_n \to \theta^*\} \bigcap \Big\{ \bigcup_{m=1}^{+\infty} A_{i,+\infty} \Big\} \bigg)$$

$$= P\bigg( \bigcup_{i=1}^{+\infty} \{\theta_n \to \theta^*\} \bigcap A_{i,+\infty} \bigg)$$

$$= P\bigg( \bigcup_{i=1}^{+\infty} \Big\{ \lim_{n\to+\infty} I_{i,+\infty} R(\theta_n) = 0 \Big\} \bigg)$$

$$\leq \sum_{i=1}^{+\infty} P\Big( \lim_{n\to+\infty} I_{i,+\infty} R(\theta_n) = 0 \Big)$$

$$= 0 \,.$$

### A.7 PROOF OF THEOREM 3.4.

First we order $J_\infty^*$ as $\{\theta_i^*\}$. Then Assumption 2.2 implies that $\forall\, \theta^* \in J^*$, there is $\|\tilde{\nabla} g(\theta)\| > 0$ ($g$ is smooth at $\theta$ and $\theta \in U(\theta^*, \delta_{\theta^*})/\{\theta^*\}$). That means for any $\theta_i^* \neq \theta_j^* \in J^*$, there is $\|\theta_i^* - \theta_j^*\| \geq \inf_{\theta_i \neq \theta_j} \|\theta_i^* - \theta_j^*\| := \hat{\delta}_0 \neq 0$ and $U(\theta_i^*, \delta_{\theta_i^*}) \cap U(\theta_j^*, \delta_{\theta_j^*}) = \emptyset$. Furthermore, it means that there are at most infinite $\{\theta_i^*\}$. We assign this number as $m$. Due $\liminf_{\theta\to+\infty} \|\tilde{\nabla} g\| > 0$, we know $\{\delta_{\theta_i^*}\}$ is bounded. Then we construct a function $\bar{R}(\theta)$ as follow:

$$\bar{R}_{\theta_i^*}(\theta) = \|\theta - \theta_i^*\|^{r_{\theta_i^*}+1} \,.$$

Then we try to prove that there exists a function $\hat{R}(\theta)$ satisfies:

1. For any $\theta \in \mathbb{R}^d$, there exist $H_{\theta\theta}$ such that $\theta^T H_{\theta\theta}(\hat{R})\theta \leq \big( \max_{\theta_i^* \in J_\infty^*} r^{\theta_i^*}(r^{\theta_i^*} + 1) \big) \|\theta\|^2$.
2. $\hat{R}(\theta) = \|\theta - \theta_i^*\|^{r_{\theta_i^*}+1}$, when $\theta$ near the $\theta_i^*$.
3. $\hat{R}(\theta)$ is bounded.

We define indicator functions

$$\hat{I}_{\theta_i^*}^{(r_i)} := \begin{cases} 1, & \text{if } \|\theta - \theta^*\| \leq r_i \\ 0, & \text{if } \|\theta - \theta^*\| > r_i \end{cases},$$

where $r_i$ is an undetermined coefficient. Clearly, function $\hat{I}_{\theta_i^*}^{(r_i)} \bar{R}_{\theta_i^*}(\theta)$ can be seen as an unary function $f_{\theta_i^*}(x) = x^{r_{\theta_i^*}+1}$, $(0 < x < r_i)$ about the independent variable $\|\theta - \theta_i^*\|$. Then for any $\bar{\delta}_0 > 0$, we can always find

$$h_{\theta_i^*}(x) = r_i^{r_{\theta_i^*}+1} + \frac{(r_{\theta_i^*} + 1)^2 r_i^{2r_{\theta_i^*}}}{2} \,,$$

to ensure there is a smooth connection (a parabola) between $f_{\theta_i^*}(x)$ and $h_{\theta_i^*}(x)$. Denote this entirety after adding the smooth connection between $f_{\theta_i^*}(x)$ and $h_{\theta_i^*}(x)$ as $j_{\theta_i^*}(x)$, $j_{\theta_i^*}(x)$ satisfied $j''(x) < 1$ and the connection point on $h_{\theta_i^*}(x)$ is $\hat{r}_i(r_i) := r_i + (r_{\theta_i^*} + 1)r_i^{r_{\theta_i^*}}$. Then let $h_{\theta_i^*}(x)$ be an arbitrary constant value $\overline{M}$, for different $r_{\theta_i^*}$, we can always get an inverse solution $r_i := h_{\theta_i^*}^{-1}(\overline{M})$. Take $K_0 := \min_{\theta_i^* \in J_\infty^*} \{\hat{I}_{\theta_i^*}^{\hat{\delta}_{\theta_i^*}} \bar{R}_{\theta_i^*}(\theta), 1\}$, there must exists $\overline{K}_0 < K_0$, such that sets $\{U(\theta_i^*, \hat{r}_i(h_{\theta_i^*}^{-1}(\overline{K}_0)))\}$ do not intersect. Then

$$\hat{R}(\theta) := \begin{cases} \sum_{i=1}^m \hat{I}_{\theta_i^*}^{(\hat{r}_i(h_{\theta_i^*}^{-1}(\overline{K}_0)))} j_{\theta_i^*}(\|\theta - \theta_i^*\|), & \text{if } \theta \in \bigcup_{i=1}^m U(\theta_i^*, \hat{r}_i(h_{\theta_i^*}^{-1}(\overline{K}_0))) \,, \\ \overline{K}_0, & \text{others} \end{cases} \tag{35}$$

is what we need. We next discuss this problem case by case according to the value of $\hat{r}$.

The first case is $\hat{r} = 1$ (from here to equation 38), we define an event

$$A_{n,\theta_i^*}^{(\hat{l}_0)} = \{\theta_n \in U(\theta_i^*, h_{\theta_i^*}^{-1}(\overline{K}_0))\}\,,$$

and the characteristic function be $I_{n,\theta_i}^{(\hat{l}_0)}$. Then we can get that

$$
\begin{aligned}
I_{n,\theta_i^*}^{(\hat{l}_0)}\big(\hat{R}(\theta_{n+1}) - \hat{R}(\theta_n)\big) &\le -I_{n,\theta_i^*}\frac{\hat{k}_1\epsilon_0}{2}\|\tilde{\nabla}\hat{R}(\theta_n)\|^2 + \hat{\zeta}_n \\
&\le -I_{n,\theta_i^*}^{(\hat{l}_0)}\frac{k_0\hat{k}_1\epsilon_0}{2}\hat{R}(\theta_n) + I_{n,\theta_i^*}^{(\hat{l}_0)}\hat{\zeta}_n,
\end{aligned}
\tag{36}
$$

where $\{\hat{\zeta}_n\}$ is a Martingale difference sequence defined as

$$\hat{\zeta}_n := \epsilon_0\|\tilde{\nabla}\hat{R}(\theta_n)\|^2 - \tilde{\nabla}\hat{R}(\theta_n)^T v_n + 2M_0\|v_n\|^2 - 2M_0\,\mathbb{E}(\|v_n\|^2|\mathcal{F}_n)\,,$$

where $k_0$, $\hat{k}_1$ are two constants. We also define $I_n^{(-\hat{l}_0)} := \mathbf{1} - \sum_{i=1}^m I_{n,\theta_i^*}^{(\hat{l}_0)}$, and obtain

$$
\begin{aligned}
I_n^{(-\hat{l}_0)}\big(\hat{R}(\theta_{n+1}) - \hat{R}(\theta_n)\big) &\le I_n^{(-\hat{l}_0)}\overline{K}_0 \le I_n^{(-\hat{l}_0)}\hat{R}(\theta_n)\frac{\overline{K}_0}{\hat{R}(\theta_n)} \\
&\le I_n^{(-\hat{l}_0)}\hat{R}(\theta_n)\frac{1^{r_{\theta_i^*}+1} + \frac{(r_{\theta_i^*}+1)^2 1^{2r_{\theta_i^*}}}{2}}{1} \\
&\le 3I_n^{(-\hat{l}_0)}\hat{R}(\theta_n)\,.
\end{aligned}
\tag{37}
$$

Through calculating the sum of equation 36, equation 37, we obtain

$$\hat{R}(\theta_{n+1}) - \hat{R}(\theta_n) \le -\frac{k_0\hat{k}_1\epsilon_0}{2}\hat{R}(\theta_n) + 3I_n^{(-\hat{l}_0)}\hat{R}(\theta_n) + \hat{\zeta}'_n\,,$$

$$\mathbb{E}\big(\hat{R}(\theta_{n+1})\big|\mathcal{F}_n\big) \le \left(1 - \frac{k_0\hat{k}_1\epsilon_0}{2} + 3I_n^{(-\hat{l}_0)}\right)\hat{R}(\theta_n)\,,$$

where $\hat{\zeta}'_n := \sum_{i=1}^m I_{n,\theta_i^*}^{(\hat{l}_0)}\hat{\zeta}_n$. Denote $k' := k_0\hat{k}_1\epsilon_0/2$, we get

$$\mathbb{E}\left(\frac{\hat{R}(\theta_{n+1})}{\prod_{k=1}^n \big(1 - k'\epsilon_0 + 3I_k^{(-\hat{l}_0)}\big)}\bigg|\mathcal{F}_n\right) \le \frac{\hat{R}(\theta_n)}{\prod_{k=1}^{n-1}\big(1 - k'\epsilon_0 + 3I_k^{(-\hat{l}_0)}\big)}\,.$$

Through the upper martingale convergence theorem, we get

$$\hat{R}(\theta_n) = O\left(\prod_{k=1}^{n-1}\big(1 - k'\epsilon_0 + 3I_k^{(-\hat{l}_0)}\big)\right)$$

almost surely. By Theorem 3.1, we also $\sum_{k=1}^{+\infty} I_k^{(-\hat{l}_0)} < +\infty$ almost surely, which means

$$\hat{R}(\theta_n) = O\big((1 - k'\epsilon_0)^n\big)$$

almost surely. Denote

$$p_0 := 1 - k'\epsilon_0 < 1\,,$$

we have

$$\hat{R}(\theta_n) = O\big(p_0^n\big)\ \text{almost surely}\,. \tag{38}$$

The second case is when $\hat{r} > 1$. Let

$$\hat{l}_0 := \min_{1 \le i \le m,\ r_{\theta_i^*} > 1}\left\{\min\left\{\left(\frac{\beta_{\theta_*} + 1}{2r(r+1)G_0^{(r_{\theta^*})}\alpha_{\theta^*}\epsilon_0^r}\right)^{\frac{r_{\theta_i^*}+1}{r_{\theta_i^*}-1}}, \delta_{\theta_i^*}, h_{\theta_i^*}^{-1}(\overline{K}_0)),\right\}\right\}\,,$$

Then we construct a set

$$S_{\theta_i^*}^{(\hat{l}_0)} = \{\theta\big|0 \le \|\theta - \theta_i^*\| < \hat{l}_0\,.$$

We also define event $A_{n,\theta_i^*}^{(\hat{l}_0)} = \{\theta_n \in S^{(\hat{l}_0)}\}$ and let the characteristic function be $I_{n,\theta_i}^{(\hat{l}_0)}$. Then we can get that

$$
\begin{aligned}
I_{n,\theta_i^*}^{(\hat{l}_0)}\big(\hat{R}(\theta_{n+1}) - \hat{R}(\theta_n)\big) &\leq -I_{n,\theta_i^*}\frac{\hat{k}_1\epsilon_0}{2}\|\tilde{\nabla}\hat{R}(\theta_n)\|^2 + \hat{\zeta}_n \\
&\leq -I_{n,\theta_i^*}^{(\hat{l}_0)}\frac{k_0\hat{k}_1\epsilon_0}{2}\hat{R}^{\frac{\hat{r}+1}{2}}(\theta_n) + I_{n,\theta_i^*}^{(\hat{l}_0)}\hat{\zeta}_n ,
\end{aligned}
\tag{39}
$$

where $\{\hat{\zeta}_n\}$ is a Martingale difference sequence defined as

$$
\hat{\zeta}_n := \epsilon_0\|\tilde{\nabla}\hat{R}(\theta_n)\|^2 - \tilde{\nabla}\hat{R}(\theta_n)^T v_n + 2M_0\|v_n\|^{r+1} - 2M_0\,\mathbb{E}(\|v_n\|^{r+1}|\mathcal{F}_n) ,
$$

and $k_0$, $\hat{k}_1$ are two constants. Define $I_n^{(-\hat{l}_0)} := \mathbf{1} - \sum_{i=1}^m I_{n,\theta_i^*}^{(\hat{l}_0)}$, we get

$$
I_n^{(-\hat{l}_0)}\big(\hat{R}(\theta_{n+1}) - \hat{R}(\theta_n)\big) \leq I_n^{(-\hat{l}_0)}\overline{K}_0 \leq \hat{a}_0 I_n^{(-\hat{l}_0)}\hat{R}^{\frac{\hat{r}+1}{2}}(\theta_n),
\tag{40}
$$

where $\hat{a}_0$ is a constant. Through calculating the sum of equation 39 and equation 40, we get

$$
\hat{R}(\theta_{n+1}) - \hat{R}(\theta_n) \leq -\frac{k_0\hat{k}_1\epsilon_0}{2}\hat{R}^{\frac{\hat{r}+1}{2}}(\theta_n) + I_n^{(-\hat{l}_0)}\hat{a}_0\hat{R}^{\frac{\hat{r}+1}{2}} + \hat{\zeta}'_n,
$$

where $\hat{\zeta}'_n := \sum_{i=1}^m I_{n,\theta_i^*}^{(\hat{l}_0)}\hat{\zeta}_n$. We also have

$$
\hat{R}(\theta_{n+1}) \leq \hat{R}(\theta_n)\left(1 - k'\hat{R}^{\frac{\hat{r}-1}{2}}(\theta_n) + I_n^{(-\hat{l}_0)}\hat{a}_0\hat{R}^{\frac{\hat{r}-1}{2}}(\theta_n) + \frac{\hat{\zeta}'_n}{\hat{R}(\theta_n)}\right).
$$

This

$$
\hat{R}^{\frac{1-\hat{r}}{2}}(\theta_{n+1}) \geq \hat{R}^{\frac{1-\hat{r}}{2}}(\theta_n)\left(1 - k'\hat{R}^{\frac{\hat{r}-1}{2}}(\theta_n) + I_n^{(-\hat{l}_0)}\hat{a}_0\hat{R}^{\frac{\hat{r}-1}{2}}(\theta_n) + \frac{\hat{\zeta}_n}{\hat{R}(\theta_n)}\right)^{\frac{1-\hat{r}}{2}}.
$$

Using the inequalities $(1+x)^{r_0} \geq 1 + r_0 x,\ (r_0 < 0)$, we have

$$
\hat{R}^{\frac{1-\hat{r}}{2}}(\theta_{n+1}) \geq \hat{R}^{\frac{1-\hat{r}}{2}}(\theta_n) + \frac{k'(\hat{r}-1)}{2} + \frac{(1-r)}{2}I_n^{(-\hat{l}_0)}\hat{a}_0 + \frac{(1-r)\hat{\zeta}_n}{2\hat{R}^{\frac{\hat{r}+1}{2}}(\theta_n)}.
$$

Summing this over $n$, we have

$$
\hat{R}^{\frac{1-\hat{r}}{2}}(\theta_{n+1}) \geq \hat{R}^{\frac{1-\hat{r}}{2}}(\theta_1) + \frac{k'(\hat{r}-1)}{2}n + (1-\hat{r})\hat{a}_0\sum_{k=1}^n I_k^{(-\hat{l}_0)} + \sum_{k=1}^n\frac{(1-\hat{r})\hat{\zeta}_k}{2\hat{R}^{\frac{\hat{r}+1}{2}}(\theta_k)}.
$$

Note that $\sum_{k=1}^{+\infty} I_k^{(-\hat{l}_0)} < +\infty$ almost surely, thus we have

$$
\hat{R}^{\frac{1-\hat{r}}{2}}(\theta_{n+1}) \geq \Omega(n) + \sum_{k=1}^n\frac{(1-\hat{r})\hat{\zeta}_k}{2\hat{R}^{\frac{\hat{r}+1}{2}}(\theta_k)} \text{, almost surely}.
$$

Denote

$$
\hat{\zeta}'_n := \frac{(1-r)\hat{\zeta}_n}{2\hat{R}^{\frac{\hat{r}+1}{2}}(\theta_n)}.
$$

Clearly,

$$
\sup_n \mathbb{E}\left(\|\hat{\zeta}'_n\|^2|\mathcal{F}_n\right) = \frac{(r-1)^2}{4}\sup_n \mathbb{E}\left(\left\|\frac{\hat{\zeta}_k}{\hat{R}^{\frac{\hat{r}+1}{2}}(\theta_k)}\right\|^2\Big|\mathcal{F}_n\right) < +\infty \text{almost surely}.
$$

By Lemma A.1, we have

$$
\sum_{k=1}^n\frac{(1-\hat{r})\hat{\zeta}_k}{2\hat{R}^{\frac{\hat{r}+1}{2}}(\theta_k)} = O(\sqrt{n}\ln(n))\text{almost surely}.
$$

Then,

$$
\hat{R}^{\frac{1-\hat{r}}{2}}(\theta_n) \geq \Omega(n)\text{almost surely},
$$

which implies

$$
g(\theta_n) = O\big(\hat{R}(\theta_n)\big) = O\big(n^{-\frac{2}{\hat{r}-1}}\big)\text{almost surely}.
$$

## A.8 Proof of Corollary 3.1

When the loss function $g(\theta_n)$ attains the $\varepsilon'$ accuracy, according to Theorem 3.4,the overall number of SGD iteration is

$$
n = \begin{cases} O\big(\log(\frac{1}{\varepsilon'})\big) & \text{almost surely}, \quad \text{if } \hat{r} = 1 \\ O\big((\frac{1}{\varepsilon'})^{\frac{\hat{r}-1}{2}}\big) & \text{almost surely}, \quad \text{if } \hat{r} > 1. \end{cases}
$$

Then we consider the computational time of a single step of SGD. Generally, the main time-consuming part of one step is computing the gradient of loss function on a batch of datasets, which can be decomposed into computing $N_0$ times of numerical differentiation, where the $N_0$ is the size of the dataset. We assume time consumed of computing a function value is $O(1)$. When a specific numerical differentiation scheme is given, such as $\frac{\partial f(\theta^{(1)},\cdots,\theta^{(d)},x)}{\partial \theta_i}\big|_{\theta=\theta_0} \approx \frac{f(\theta_0^{(1)},\cdots,\theta_0^{(i)}+h,\cdots,\theta_0^{(d)},x)-f(\theta_0,x)}{h}$, it's obviously the computation time of numerical gradient is $O(d)$. In summary, the whole computation time is

$$
\begin{cases} O\big(N_0 d \cdot \log(\frac{1}{\varepsilon'})\big) & \text{almost surely}, \quad \text{if } \hat{r} = 1 \\ O\big(N_0 d \cdot (\frac{1}{\varepsilon'})^{\frac{\hat{r}-1}{2}}\big) & \text{almost surely}, \quad \text{if } \hat{r} > 1, \end{cases}
$$

which is bounded by a polynomial time.

