# OpenReview forum: "On the convergence of SGD under the over-parameter setting"
_ICLR.cc/2023/Conference — Submitted to ICLR 2023_

### Official Review · Reviewer_fVFB · 2022-10-19

**Confidence:** 5
**Clarity, Quality, Novelty And Reproducibility:** The writing can be improved, some sen…
**Correctness:** 1
**Technical Novelty And Significance:** 2
**Empirical Novelty And Significance:** Not applicable
**Recommendation:** 1

**Strength And Weaknesses:**

I have a few technical concerns on different aspects of the theoretical results in this paper.

Problematic assumptions:

Assumption 1: $g(\theta)$ is already non-negative by definition, and there is no need for repitition. The actual meaning of smoothness needs to be made clear here ($C^1$ or $C^{\infty}$ ?)

Assumption 3: the set of roots itself is already a closed set, as long as $g(\theta)$ is continuous which is implicit: one only needs to prove any limit point $x$ of roots $\{x_i\}$ is still a root. Suppose for contradiction $g(x)>0$, then by its continuity there exists an open neighborhood $S$ containing $x$, such that for any $x'\in S$ we have that $g(x')>g(x)/2>0$. However, some root $x_N$ will fall into $S$ which finishes the proof.

Assumption 4: this is a rather strong assumption in my opinion. The simple example $f(\theta,x)=(\theta-x)^2$ fails to satisfy the assumption, let alone neural networks. Even if the domain is restricted to be a compact set of $\mathbb{R}^d$, the constant $c$ is likely to be a large poly factor in the dimension $d$.

Assumption 2.3: can you give any example included in the large family of functions as claimed?

$\tilde{\nabla}g(\theta)$ is not defined. Is it an unbiased estimator of $\nabla g(\theta)$ or what? The model of SGD isn't explained either.

Main results:

I will only take Theorem 3.1 as an example. The argument in step 3 is doubtful: $\theta_n$ might not converge at all with positive probability. Suppose I am the adversary that generates the stochastic estimator $\tilde{\nabla}g(\theta)$, then at any root I'm allowed to generate $\pm a v$ by Assumption 2.2 where $v$ is randomly drawn from the unit sphere. Then such sequence of $\{\theta_n\}$ doesn't converge, and one can only talk about limit points. Unfortunately, even a non-root point can be a limit point if $\tilde{\nabla}g(\theta)$ is an unbiased estimator.

I suspect that the proof goes through when $\tilde{\nabla}g(\theta)$ is exact and the randomness of SGD decreases with function value, as in equation (5). However, this is a very strong and non-standard assumption and I feel it's not what the authors mean.


**Summary Of The Paper:**

This paper claims that SGD converges to global minimum almost surely, under some assumptions.

**Summary Of The Review:**

Based on the mathematical issues I found, I doubt the correctness of this theoretical paper.

---

> ### Author Response · Authors · 2022-11-07
> **Response**
>
> Thank you for your review. Firstly, we apologize for missing the definition of the $\tilde{\nabla}$ at the beginning of the paper. Specifically, $\tilde{\nabla}$ means a randomly chosen value from the $\partial g$ (due to $\partial g$ is actually a set, we can not calculate that directly. We add the definition of  $\tilde{\nabla}$ in Definition 1 of the revisited version. We provide detailed responses to your questions.
>
> Assumption 1: $g(\theta)$ is already non-negative by definition, and there is no need for repitition. The actual meaning of smoothness needs to be made clear here ($C^1$ or $C^{\infty}$).
>
> Answer:
> In our paper, we mainly consider a series of problems that is continuous (in $C$) but not smooth (in $C\setminus C^1$). To be exact, we assume that the loss function is smooth almost everywhere (the measure of the non-smooth point is 0): The loss function $g(\theta)$ is continuously differentiable in $\mathbb{R}^{d}/\Omega,$ where $\Omega$ is a set with $0$ Lebesgue measure. A typical example $f(x)=\min\\{x^{2},(x-1)^{2}\\}.$
>
>
>
> Assumption 3:
> The set of roots itself is already a closed set, as long as $g(\theta)$ is continuous which is implicit: one only needs to prove any limit point $x$ of roots $x_i$ is still a root. Suppose for contradiction $g(x)>0$, then by its continuity there exists an open neighborhood $S$ containing $x$, such that for any $x′\in S$ we have that $g(x′)>g(x)/2>0$. However, some root $x_N$ will fall into $S$ which finishes the proof.
>
> Answer:
> We made a typo in the original assumption. The assumption is that $J^\ast$ can be written as the union of countably many *connected components* (instead of closed sets). The definition of a connected component is: $J_i$ is called a connected component of $J^*$ if it is a connected subset of $J^*$ and any subset $\hat{J}\supset J_i$ is not a connected set. The purpose that we consider the connected component is the loss function $g(\theta)$ has the same value in it.
>
>
>
>
> Assumption 4:
> This is a rather strong assumption in my opinion. The simple example $f(\theta,x)=(\theta-x)^2$ fails to satisfy the assumption, let alone neural networks. Even if the domain is restricted to be a compact set of $R^d$, the constant $c$ is likely to be a large poly factor in the dimension $d$.
>
> Answer:
> First, this function is a continuous differentiable function. That means its subgradient $\tilde{\nabla} f(\theta,x)$ is its gradient $\nabla f(\theta,x).$ We can write the concrete form of subgradient $\tilde{\nabla} f(\theta,x)=2(\theta-x).$ Then we can find for any $\theta_1$ and $\theta_2,$ there is $\\|\tilde{\nabla}f(\theta_1,x)-\tilde{\nabla}f(\theta_2,x)\\|=2\\|\theta_1-\theta_2\\|\le 2\max\\{\\|\theta_1-\theta_2\\|,1\\}.$ Then we have $\liminf_{\theta\rightarrow\infty}\\|\tilde{\nabla}f(\theta,x)\\|=\liminf_{\theta\rightarrow\infty}2\\|\theta-x\\|=+\infty.$ Naturally, $\liminf_{\theta\to \infty}\\|\tilde{\nabla}f(\theta,x)\\|>\\{4c\sqrt{M_{0}},4c\sqrt{K_{0}}\\}.$
> In fact, the first part of this assumption is a non-smooth extension of the normal assumption 'L-smooth assumption', and the second part for MSE loss function and the over-parameter setting is also not strong.
>
> Assumption 2.3 (Assumption 2.2 in the updated manuscript):
> Can you give any example included in the large family of functions as claimed?
>
> Answer
>
> For example
> $g(\theta)=\\|\theta-\theta_{1}\\|^{2}$, $g(\theta)=\min\\{\\|\theta-\theta_{1}\\|^{2},\\|\theta-\theta_{2}\\|^{2}\\},$ $g(\theta)=\min\\{\\|\theta\\|^{2},\\|\theta\\|^{3}\\}$ are under our assumptions.
>
> Question 1:
> $\tilde{\nabla}g(\theta)$ is not defined. Is it an unbiased estimator of $\nabla g(θ)$ or what? The model of SGD isn't explained either.
>
> Answer
>
> This $\tilde{\nabla} g$ is not an unbiased estimate about the gradient. It is a randomly chosen value of the Clark subdifferenial $\partial g.$ In the new version of the manuscript it is provided in Definition 1.

---

> > ### Author Response · Authors · 2022-11-07
> > **Response**
> >
> > Question on main results:
> >
> > > I will only take Theorem 3.1 as an example. The argument in step 3 is doubtful: $\theta_n$ might not converge at all with positive probability. Suppose I am the adversary that generates the stochastic estimator
> > $\tilde{\nabla}g$, then at any root I'm allowed to generate $\pm av$ by Assumption 2.2 where $v$ is randomly drawn from the unit sphere. Then such sequence of $\theta_{n}$ doesn't converge, and one can only talk about limit points. Unfortunately, even a non-root point can be a limit point if
> > $\tilde{\nabla}g$ is an unbiased estimator.
> > I suspect that the proof goes through when
> > $\tilde{\nabla}g$ is exact and the randomness of SGD decreases with function value, as in equation (5). However, this is a very strong and non-standard assumption and I feel it's not what the authors mean.
> >
> > Response:
> > In our paper, we only consider two types of noise, the original sampling noise of SGD and the sampling noise with global convergence guarantee. No matter which type we consider, the noise satisfies the variance is $0$ at $J^*$. For sampling noise, from Claim 2.1, we get $E_{\xi_{n}}\\|\tilde{\nabla}g(\theta^{\ast},\xi_{n})\\|^{2}=0$, (for $g(\theta^{\ast})=0$). For the sampling noise with global convergence guarantee, we get $E_{\xi_{n}}\\|v_{n}\\|^{2}$=$E_{\xi_{n}}\\|\tilde{\nabla}g(\theta^{\ast},\xi_{n})\\|^{2}+\hat{k}\min\\{g(\theta^{\ast}),K_{0}\\}=0$ (when $\theta_{n}=\theta^{\ast}$). No matter which type the noise holds. This stochastic dynamical systems will be absolutely static when $\theta$ reaches the global optimum. We are not investigating the particular kind of noise you mentioned in the review.

---

> > > ### Author Response · Authors · 2022-11-11
> > > **Response**
> > >
> > > Dear review fvFB:
> > >
> > > We did not receive your response since we submitted rebuttals. Did you receive our rebuttal?
> > > Best regards,
> > >
> > > Authors of Paper6137

---

> > > > ### Author Response · Authors · 2022-11-15
> > > > **Concern**
> > > >
> > > > Dear reviewer fvFB:
> > > >
> > > > Did you see $\max$ which is in Assumption 2.1 item 4 as $\min?$

---

> > > > > ### Author Response · Authors · 2022-11-16
> > > > > **Concern**
> > > > >
> > > > > Dear reviewer fvFB:
> > > > >
> > > > > Can you  please take a look at our responses and reply to us? Questions you pointed out  have many mistakes. Can you communicate with us?
> > > > >
> > > > > authors

---

> > > > > > ### Author Response · Authors · 2022-11-17
> > > > > > **Concern (again)**
> > > > > >
> > > > > > Dear reviewer fvFB:
> > > > > >
> > > > > >
> > > > > > Can you please take a look at our responses and reply to us? Questions you pointed out have many mistakes. Can you communicate with us?
> > > > > >
> > > > > >
> > > > > > authors

---

> > > > > > > ### Author Response · Authors · 2022-11-17
> > > > > > > **Concern(again again)**
> > > > > > >
> > > > > > > Dear reviewer fvFB:
> > > > > > >
> > > > > > > Can you please take a look at our responses and reply to us? Questions you pointed out have many mistakes. Can you communicate with us?
> > > > > > >
> > > > > > > authors

---

### Official Review · Reviewer_XXfc · 2022-10-24

**Confidence:** 1
**Clarity, Quality, Novelty And Reproducibility:** Paper is written well, well-organized…
**Correctness:** 4
**Technical Novelty And Significance:** 3
**Empirical Novelty And Significance:** Not applicable
**Recommendation:** 6

**Strength And Weaknesses:**

Strength:
The paper is technically sound.
The paper studies an interesting question related to the over-parameterized problem and provides more understanding of the optimization landscape.
The theoretical results are important to the machine learning community.

Weakness；
More discussion should be added after each Theorem.


**Summary Of The Paper:**

This paper studies the convergence property of SGD under the over-parameter setting. Under some regular assumptions of the loss and subdifferential gradient noise, this paper gives some novel results: (1) SGD must converge to a global optimum with probability 1. (2) SGD converges to a sharper global optimum not as easy as a flat one. (3) SGD achieves an arbitrary accuracy in polynomial time.

**Summary Of The Review:**

The paper provides some novel insights on SGD’s optimization landscape. For example, the authors show interesting and important findings that SGD could indeed obtain a global optimum even in the non-smooth non-convex over-parameter setting.
I have some concerns.
(1) This paper studies square loss. Can these results be generalized to general losses?
(2) What’s the difference between $\tilde{\nabla}$ and $\nabla$. subdifferential?
(3) Can the authors provide some numerical experiments to support the theory in this paper?

---

> ### Author Response · Authors · 2022-11-07
> **Response**
>
> Question 1:
>
> This paper studies square loss. Can these results be generalized to general losses?
>
> Answer:
>
> We believe that it also generalizes to the cross entropy loss and we are working on that.
>
>
> Question 2:
>
> Question on the definition of the subdifferential.
>
> Answer:
>
> We wanted to clarify that $\tilde{\nabla}$ means a randomly chosen value from the Clarke subdifferential $\partial g$. We apologize for missing this definition in the original submission and we have fixed that in the revised version.
>
> Question 3:
>
> Can the authors provide some numerical experiments to support the theory in this paper?
>
> Answer:
>
> The manuscript's main focus is on insights about SGD - an algorithm that are already known to work very well in practice. The manuscripts states that when overparametrized, SGD indeed enjoys global convergence. Other than SGD itself, we do not have much to test in the experiments, and SGD has obviously been tested extensively so we do not really have much to contribute in the empirical sense.

---

### Official Review · Reviewer_zzGL · 2022-10-24

**Confidence:** 3
**Clarity, Quality, Novelty And Reproducibility:** See the above comments.
**Correctness:** 3
**Technical Novelty And Significance:** 3
**Empirical Novelty And Significance:** Not applicable
**Recommendation:** 5

**Strength And Weaknesses:**

Strength:
This paper considers two fundamental problems in learning, i.e., does SGD provably find the global optimum with an over-parametrized model, and why SGD prefers a flat global minimum? If all the claims (Thm 3.1, 3.2, 3.3) in this paper are correct, it is indeed a breakthrough in this field.

Weaknesses:
However, it is hard for me to judge the correctness of the draft due to the following writing issue.

1. There are so many notations used without definition. For example, $\tilde{\nabla}$ in assumption 2.1 (it means subdifferential?), $N_0$ in the proof of Claim 2.2 (number of samples in a batch?), $K_0$ in equation (5) and $\zeta_n$ in equation (9).

2. I did not see the importance of Claims 2.1 and 2.2, but they took up much space in the paper. Moreover, the sketch of Thm 3.1 seems to be disconnected from the preliminaries in section 2.

I would recommend the authors add more insights and intuitions of the proof instead of the proof details and provide some high-level comparisons on why the proposed techniques could show such strong results. This paper considers two fundamental problems in learning, i.e., does SGD provably find the global optimum with an over-parametrized model, and why SGD prefers a flat global minimum? If all the claims (Thm 3.1, 3.2, 3.3) in this paper are correct, it is a breakthrough in this field.

Some related references are missing. I would like the authors to address how their approach and results differ from these papers.

Vaswani, Sharan, Francis Bach, and Mark Schmidt. "Fast and faster convergence of sgd for over-parameterized models and an accelerated perceptron." In The 22nd international conference on artificial intelligence and statistics, pp. 1195-1204. PMLR, 2019.

Chizat, Lenaic, and Francis Bach. "On the global convergence of gradient descent for over-parameterized models using optimal transport." Advances in neural information processing systems 31 (2018).


**Summary Of The Paper:**

This paper utilizes a method based on Clarke subdifferential and Lyapunov stability to show that SGD for an over-parametrized model converges to the global optimum almost surely under arbitrary initial value and some mild assumptions on the loss function. In addition, it is also shown that if the learning rate is larger than a value depending on the structure of a global minimum, the probability of converging to this global optimum is zero.

**Summary Of The Review:**

This paper considers a fundamental problem in the convergence guarantee of SGD in an over-parametrized model. However, the paper is not well-written in its current form, which makes the reviewer hard to verify its correctness.

---

> ### Author Response · Authors · 2022-11-07
> **Response**
>
>
>
> We thank the reviewer for their insightful comments and constructive feedback. We believe that the concerns in the review are minor issues and could be addressed by providing the below responses.
>
> We first wanted to clarify that $\tilde{\nabla}$ means a randomly chosen value from the Clarke subdifferential $\partial g$. We apologize for missing this definition in the original submission and we have fixed that in the revised version.
>
> Question 1:There are so many notations used without definition. For example, $\tilde{\nabla}$
>  in assumption 2.1 (it means subdifferential?)......
>
> Answer:
> $\tilde{\nabla}$ means an randomly chosen value from the Clarke subdifferential $\partial g$. $N_{0}$ is the mini-batch size and $K_{0}$ is used to limit the size of the Gaussian noise (can take any finite value). All notation are now properly defined in the updated manuscript.
>
> Question 2:
>
> I did not see the importance of Claims 2.1 and 2.2, but they took up much space in the paper. Moreover, the sketch of Thm 3.1 seems to be disconnected from the preliminaries in Section 2.
>
> Answer:
> Claim 2.1 make the variance of the sampling noise $E_{\xi_{n}}\\|\tilde{\nabla}g(\theta,\xi_{n})-\tilde{\nabla}g(\theta)\\|^{2}=0 $ when $\theta$ reach the global optimum, which means once the $\theta_{n}$ reach some global optimum, it will not escape from it. This is the key observation to establish a global convergence in the over-parametrized setting. Claim 2.2 states that the loss function is smooth at the global optimum. Both claims are the key observations obtained under the over-parametrized setting. Otherwise without these two claims it becomes a general non-smooth non-convex optimization problem, and such gloabl convergence results would not be anywhere possible.
>
>
>
> Question 3:
> I would recommend the authors add more insights and intuitions of the proof instead of the proof details and provide some high-level comparisons on why the proposed techniques could show such strong results. This paper considers two fundamental problems in learning, i.e., does SGD provably find the global optimum with an over-parametrized model, and why SGD prefers a flat global minimum? If all the claims (Thm 3.1, 3.2, 3.3) in this paper are correct, it is a breakthrough in this field.
>
>
> Answer:
>
> The basic intuition is as follows. We first understand the SGD as a Markov chain with the continuous state space. Then we aim to prove that the global optimum is the only absorbing state of this Markov chain. Concretely, due to the property of the sampling noise, this noise enjoys 0  variance when $\theta$ reaches the global optimum (Claim 2.1), i.e., $E_{\xi_n}\\|\tilde{\nabla}g(\theta,\xi_n)-\tilde{\nabla}g(\theta)\\|^{2}=0,$ which guarantees that once $\theta_n$ reaches the global optimum, it will not escape from the optimum. Meanwhile, in other local optimums, the positive variance makes $\theta_n$ jump out to this local optimum. Otherwise, due to this Markov chain is a continuous state space Markov chain, an absorbing state with the measure 0 cannot become the real absorbing state (the probability of the $\theta_n$ reaching this absorbing state in every epoch is 0). Based on this, we need this absorbing state to have a flat-enough neighborhood (Assumption 2.2 in the new version), which deduces that $\theta_n$ that fall on this neighborhood tend to move closer to this absorbing state. Combining this absorbing state and this neighborhood statement, we can prove the distribution of $\theta_n$ will concentrate on the global optimum when as the iteration goes. Finally, this distribution will degenerate to the global optimum, that is, $\theta_n$ will converge to the global optimum.
>
> This neighborhood is the key insight of proving the convergence of SGD. The neighborhood cannot be very sharp (have at most quadratic growth), which is the reason we made Assumption 2.2, item 1. It is actually reflected in Equation (8). A flat enough neighborhood can make the coefficient of the third term of (8) negative, which in turn makes the $R(\theta_n)$ (the Lyapunov function) to decrease with high probability ($\theta_n$ close to global optimum). Otherwise, if the neighborhood is sharp, this coefficient will become positive, which makes $R(\theta_n)$ increasing ($\theta_n$ away from global optimum).
>
> We have added these insights to the end of the introduction in the updated manuscript.

---

> > ### Author Response · Authors · 2022-11-07
> > **Response**
> >
> > Question 4:
> >
> > Some related references are missing. I would like the authors to address how their approach and results differ from these papers.
> >
> > Vaswani, Sharan, Francis Bach, and Mark Schmidt. "Fast and faster convergence of sgd for over-parameterized models and an accelerated perceptron." In The 22nd international conference on artificial intelligence and statistics, pp. 1195-1204. PMLR, 2019.
> >
> > Chizat, Lenaic, and Francis Bach. "On the global convergence of gradient descent for over-parameterized models using optimal transport." Advances in neural information processing systems 31 (2018).
> >
> > Answer:
> >
> > We have added the discussion on these references.

---

> ### Author Response · Authors · 2022-11-12
> **Response**
>
> Dear Reviewer zzGL,
>
>   First of all, we would like to thank you for your valuable and constructive comments which have helped improve our work substantially.
>
>  In the revised paper,  we have addressed your comments and concerns. If you think the paper has been improved, we would appreciate it if you can re-evalutate the work.   We would be happy to communicate with you!
>
>
>  Thanks a lot for your time and consideration!
>
> Best regards,
> Authors of Paper 6137

---

### Official Review · Reviewer_5yze · 2022-10-26

**Confidence:** 4
**Correctness:** 3
**Technical Novelty And Significance:** 3
**Empirical Novelty And Significance:** Not applicable
**Recommendation:** 5

**Clarity, Quality, Novelty And Reproducibility:**

The results of this paper seem to be new and the approach is different from prior work. However, its clarity and presentation is not good.

**Strength And Weaknesses:**

The strengths of this paper are the theoretical results for the global convergence of SGD almost surely. The authors consider the regular sampling scheme and propose a new scheme (sampling noise with global stable guarantee). They prove the asymptotic convergence for SGD under a set of assumptions.

The weaknesses of this paper are:
- This paper did not explain the intuition why SGD converge globally very well. The assumptions are not clearly motivated. The authors should explain why they have two set of assumptions on the gradient/ sample gradient of g. One is Assumption 2.1 part 4, line 2 where there is a lower bound on the liminf of gradient, the other is Assumption 2.2 where we put an upper bound on the sample gradient. Please add a discussion why the theory needs these assumptions and make sure they do not contradict each other. In addition, Assumption 2.3 is not natural when it asks that the constants $c_\theta, \hat{c}_\theta$ are bounded away from 0 and by a constant of $\theta$.
- The presentation of this paper is poor. There are many notations and variables that were mentioned before the authors define them in the draft. For example: $M_0$ and $a$ were referred in Assumption 2.1 but only defined until 2.2, $\tilde{\nabla} g$ has no definition, 'global stable guarantee' was referred before the explanation,... Most of the time, the sketch proofs are confusing and they did not help to understand the thought process to prove the theorems.

Question: In Theorem 3.4, what is 'the variant of Assumption 2.3 described immediately preceding this statement'?

**Summary Of The Paper:**

This paper provides theoretical results for the asymptotic convergence of SGD algorithm under an over-parameterized setting. It shows a set of assumptions that can guarantee the global convergence of SGD almost surely in some non-convex setting.

**Summary Of The Review:**

Although this paper show interesting results, I am not fully convinced by the assumptions and the intuition/reasoning behind the proofs.

---

> ### Author Response · Authors · 2022-11-07
> **Respones**
>
> \textbf{first 5 point}
>
> We thank the reviewer for their insightful comments and constructive feedback. We believe that the concerns in the review are minor issues and could be addressed by providing the below responses.
>
> We first wanted to clarify that $\tilde{\nabla}$ means a randomly chosen value from the Clarke subdifferential $\partial g$. We apologize for missing this definition in the original submission and we have fixed that in the revised version.
>
>
>
>
> Question 1: This paper did not explain the intuition why SGD converge globally very well. The assumptions are not clearly motivated. The authors should explain why they have two set of assumptions on the gradient sample gradient of $g$. One is Assumption 2.1 part 4, line 2 where there is a lower bound on the $\liminf$ of gradient, the other is Assumption 2.2 where we put an upper bound on the sample gradient. Please add a discussion why the theory needs these assumptions and make sure they do not contradict each other. In addition, Assumption 2.3 is not natural when it asks that some constants....
>
> Answer:
> First we will introduce our intuition. The basic intuition is as follows. We first understand the SGD as a Markov chain with the continuous state space. Then we aim to prove that the global optimum is the only absorbing state of this Markov chain. Concretely, due to the property of the sampling noise, this noise enjoys 0 variance when the optimization variable $\theta$ reaches the global optimum (Claim 2.1), i.e., $E_{\xi_n}\\|\tilde{\nabla}g(\theta,\xi_n)-\tilde{\nabla}g(\theta)\\|^{2}=0$ (notations are defined in the next section), which guarantees that once $\theta_n$ reaches the global optimum, it will not escape from the optimum. Meanwhile, in other local optimums, the positive variance makes $\theta_n$ jump out to this local optimum. Otherwise, as this Markov chain is a continuous state space Markov chain, an absorbing state with the measure 0 cannot become the real absorbing state (the probability of the $\theta_n$ reaching this absorbing state in every epoch is 0). Based on this, we need this absorbing state to have a flat-enough neighborhood (Assumption 2.2 in the new version), which deduces that $\theta_n$ that fall on this neighborhood tend to move closer to this absorbing state. Combining this absorbing state and this neighborhood statement, we can prove the distribution of $\theta_n$ will concentrate on the global optimum when as the iteration goes. Finally, this distribution will degenerate to the global optimum, that is, $\theta_n$ will converge to the global optimum.
>
> This neighborhood is the key insight of proving the convergence of SGD. The neighborhood cannot be very sharp (have at most quadratic growth), which is the reason we made Assumption 2.2, item 1. It is actually reflected in Equation (8). A flat enough neighborhood can make the coefficient of the third term of (8) negative, which in turn makes the $R(\theta_n)$ (the Lyapunov function) to decrease with high probability ($\theta_n$ close to global optimal). Otherwise, if the neighborhood is sharp, this coefficient will become positive, which makes $R(\theta_n)$ increasing ($\theta_n$ away from global optimal).
>
> Second, We will explain why we consider two types of noise. For the ordinary sampling noise Equation (2), $\theta_{n}$ actually converge to $J^{\ast\ast}$ (definition can reference 14 line in page 6 of our revisited version), and the global optimum $J^{\ast}$ is actually a subset of $J^{\ast\ast}.$ Much as in the large scale data set, the model $f(\theta,x)$ is complex enough to make sure that other stationary points are sensitive to the mini-batch batch selection, we still consider a global guarantee noise Equation (5) to make sure even in some extreme situation $ J^{\ast\ast}/J^{\ast}\neq \emptyset,$ SGD can converge to global optimum $J^{\ast}.$
>
>
> Third, we combine these two bounds of subgradient as one new Assumption 2.2 and change the lower bound as $\max\\{4c\sqrt{M_{0}},4c\sqrt{K_{0}}\\}$ . Next we will explain these two assumptions are not contradictory. We can calculate the lower bound (lb) as $\liminf_{\theta\rightarrow\infty}lb=\max\\{4c\sqrt{M_{0}},4c\sqrt{K_{0}}\\},$ and the upper bound (ub) as $\liminf_{\theta\rightarrow\infty}ub=M_0\liminf_{\theta\rightarrow\infty}\\|\tilde{\nabla}g(\theta)\\|^{2}.$ We can find that this assumption implies $M_{0}>1,$ and then there is $\liminf_{\theta\rightarrow\infty}ub>\lim_{\theta\rightarrow\infty}M_0\max\\{4c\sqrt{M_{0}},4c\sqrt{K_{0}}\\}>\max\\{4c\sqrt{M_{0}},4c\sqrt{K_{0}}\\}=\liminf_{\theta\rightarrow\infty}lb.$

---

> > ### Author Response · Authors · 2022-11-07
> > **Response**
> >
> > Finally, we explain Assumption 2.2 (in the previous version Assumption 2.3). The first item of this assumption is very mild. Due to Claim 2.1, we know $g(\theta)$ is smooth in $\theta^{\ast}$, that is, $\lim_{\theta\rightarrow\theta^{\ast}}\tilde{\nabla}g(\theta)=\nabla g(\theta^{\ast})=0.$ Then item 1 is to bound the speed of subgradient tend to $0$ compared with a linear function (not too sharp as $O(\sqrt{\\|\theta-\theta^{\ast}\\|})$ or $O(\\|\theta-\theta^{\ast}\\|^{0.9})$). The second item of this assumption is very close to the local Kurdyka-Lojasiewicz condition, i.e. $\\|\nabla g(\theta)\\|^{2r}\ge g(\theta)-g(\theta^{\ast})\ (r\ge 1) (\theta\in U(\theta^\ast, \delta_{\theta^{\ast}}))$ which is a typical mild condition used to substitute the local Polyak-Łojasiewicz condition (item 2 and the local Kurdyka-Lojasiewicz condition are totally equivalent for an unary function).
> >
> > Question 2 The presentation of this paper is poor. There are many notations and variables that were mentioned before the authors define them in the draft. For example...
> >
> > Answer
> >
> > The definitions of all other variables are now provided in the updated manuscript. We apologize again for missing the definition of the $\tilde{\nabla}$ at the beginning of the paper.
> >
> > Question 3:
> >
> > In Theorem 3.4, what is 'the variant of Assumption 2.3 described immediately preceding this statement?
> >
> > Answer:
> >
> > This means the statement before Theorem 3.2: "To provide the convergence rate, we will need a slightly stronger version of Assumption 2.2. We need, instead of just one $\theta^{\ast}$, all $\theta^\ast$, to satisfy the order $r_{\theta^{\ast}}+1$ expansion. In this case, the supremum of the expansion order, among all optimum points, is denoted as $\hat{r}=\max_{\theta^{*}\in J_{\infty}^{\ast}}r_{\theta^{\ast}}$, where $J_{\infty}^{\ast}:=\\{\theta^{\ast}\in J^{\ast}\mid P(\theta_{n}\rightarrow \theta^{\ast})>0\\}$."

---

> ### Author Response · Authors · 2022-11-12
> **Response**
>
> Dear Reviewer 5yze,
>
>   First of all, we would like to thank you for your valuable and constructive comments which have helped improve our work substantially.
>
>  In the revised paper,  we have addressed your comments and concerns. If you think the paper has been improved, we would appreciate it if you can re-evalutate the work.   We would be happy to communicate with you!
>
>
>  Thanks a lot for your time and consideration!
>
> Best regards,
> Authors of Paper 6137

---

### Decision · Program_Chairs · 2023-01-20

**Decision:**

Reject

**Justification For Why Not Higher Score:**

- The main concern from the reviewers is that the results rely on the strong assumptions, which are quite restrictive.
- There is no empirical evidence to show that these assumptions could potentially hold in practice.

**Justification For Why Not Lower Score:**

N/A

**Metareview: Summary, Strengths And Weaknesses:**

This paper provides theoretical results for the asymptotic convergence of SGD algorithm under an over-parameterized setting. It shows a set of assumptions that can guarantee the global convergence of SGD almost surely in some non-convex setting.

Most of the reviewers do not support its publication at ICLR and I also agree with the reviewers. The main concern from the reviewers is that the results rely on the strong assumptions, which are quite restrictive. There is no empirical evidence to show that these assumptions could potentially hold in practice.

Please take the comments and suggestions from the reviewers in their detail reviews to improve the manuscript for the future submission. I would suggest the authors to consider some optimization journal venues for this paper.